# 🌟STAR: STacked AutoRegressive Scheme for Unified Multimodal Learning

## Abstract

Multimodal large language models (MLLMs) play a pivotal role in advancing the quest for general artificial intelligence. However, achieving unified target for multimodal understanding and generation remains challenging due to optimization conflicts and performance trade-offs. To effectively enhance generative performance while preserving existing comprehension capabilities, we introduce *STAR*: *a STacked AutoRegressive scheme for task-progressive unified multimodal learning*. This approach decomposes multimodal learning into multiple stages: understanding, generation, and editing. By freezing the parameters of the fundamental autoregressive (AR) model and progressively stacking isomorphic AR modules, it avoids cross-task interference while expanding the model's capabilities. Concurrently, we introduce a high-capacity VQ to enhance the granularity of image representations and employ an implicit reasoning mechanism to improve generation quality under complex conditions. Experiments demonstrate that *STAR* achieves state-of-the-art performance on GenEval (**0.91**), DPG-Bench (**87.44**), and ImgEdit (**4.34**), validating its efficacy for unified multimodal learning.

## 1 Introduction

In recent years, the rapid advancement of multimodal large language models (MLLMs) has significantly propelled the progress of artificial general intelligence (AGI) (Touvron et al., 2023; Bi et al., 2024; OpenAI, 2024a; Team et al., 2023; DeepSeek-AI et al., 2025; Yang et al., 2025). Numerous studies have focused on constructing unified models that use a single set of parameters to simultaneously handle different tasks, such as multimodal understanding and generation (Wang et al., 2024; Chen et al., 2025c; Wang et al., 2025; Liao et al., 2025; Deng et al., 2025; Xie et al., 2025; OpenAI, 2025; Chen et al., 2025b). However, these from-scratch-trained models face a critical challenge: *inherent conflicts exist between multimodal understanding and generation tasks in both optimization objectives and feature spaces.* This often results in joint training sacrificing performance in one or more domains, thereby limiting the overall capability ceiling of unified models.

Against this backdrop, a fundamental research question emerges: *Can we continuously enhance a model's image generation capabilities while fully preserving its multimodal understanding abilities?* Existing approaches, such as MetaQuery (Pan et al., 2025) and BLIP3-o (Chen et al., 2025a), adopt a warm-started adaptation paradigm, which initializes from a pre-trained multimodal understanding model and augments it with a diffusion-based generator to enhance generation while preserving image-to-text capability. Yet, these approaches typically require **constructing feature transformation bridges** between autoregressive and diffusion models or **designing complex loss functions**, significantly increasing training complexity. Thus, we face a critical challenge: *How to extend a single MLLM in the most streamlined manner possible, enabling it to progressively acquire more sophisticated multimodal capabilities without compromising existing abilities?*

To address the aforementioned challenge, we propose *STAR* (*STacked AutoRegressive Scheme for Unified Multimodal Learning*), a novel unified learning method based on stacked autoregressive (AR) paradigm that offers three key design advantages: (i) **a task-progressive training strategy**; (ii) **a stacked autoregressive model**; and (iii) **an implicit reasoning mechanism**. Firstly, the task-progressive training paradigm decomposes unified multimodal learning into an ordered curriculum: understanding, generation, and editing, while freezing the fundamental AR backbone at each extension. This staged training paradigm simultaneously shields existing comprehension capabilities

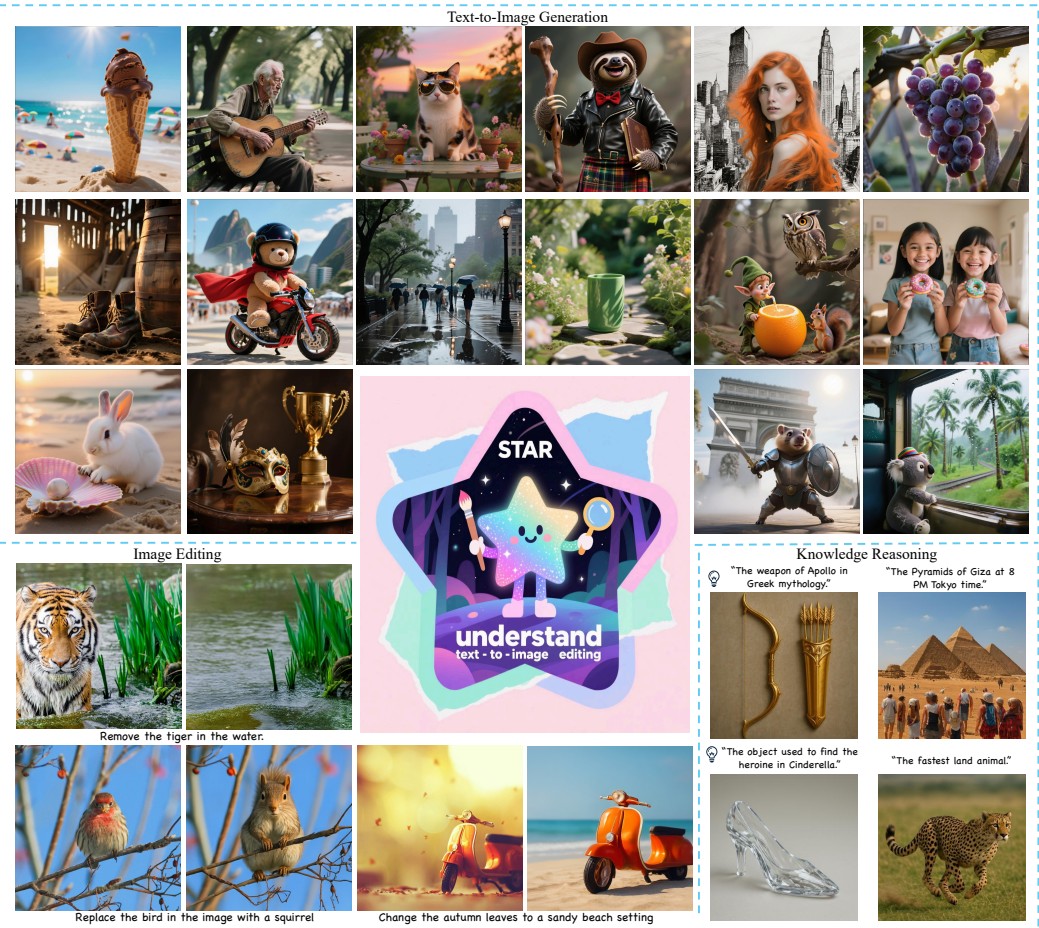

Figure 1: **STAR** enables unified multimodal learning for understanding, text-to-image, image editing, and reasoning, with a diffusion decoder enhancing the granularity of image outputs.

from catastrophic degradation and equips the model with novel generative abilities. Secondly, the stacked autoregressive model extends the frozen fundamental AR by appending a small set of isomorphic AR modules that share identical architecture and are initialized from the same parameter. The generation and editing tasks can be optimized with the standard next-token prediction objective without any auxiliary adapters or losses. This design is fundamentally distinct from MetaQuery and BLIP3-o, which rely on external bridging modules and diffusion losses. Moreover, instead of learnable queries in MetaQuery, we encode the image as discrete VQ tokens and feed them into the AR model. To furnish the token space with finer granularity, we concurrently introduce a high-capacity vector quantizer from scratch whose codebook contains *65,536* entries of *512*-d vectors, termed as **STAR-VQ**. This tokenizer is jointly optimized with a 1B-parameter and is an order of magnitude larger and denser than conventional counterparts, yielding markedly more precise visual tokens that raise the generation ceiling without codebook collapse. Finally, an implicit reasoning mechanism is introduced to harness the stacked architecture at decode time. Given a complex prompt, the fixed AR first performs inference procedure to yield implicit latent tokens, which then serve as conditional input to generate images. By explicitly separating semantic reasoning from pixel generation, this pipeline markedly improves alignment accuracy in challenging compositional and world knowledge scenarios without requiring additional parameters. Qualitative results are presented in the Figure 1.

Extensive experimental results demonstrate that the proposed **STAR** approach not only achieves leading performance across a diverse set of multimodal understanding and generation tasks, but also substantially reduces training complexity through minimal structural modifications. This highlights the advantages of progressive task expansion in unified multimodal training. We believe that **STAR** provides an insightful technical pathway toward achieving interference-free, sustainably scalable unified multimodal models. The main contributions of this work can be summarized as follows:

- We propose a task-progressive training paradigm that sequentially learns understanding, generation and editing while freezing the fundamental AR backbone, thereby safeguarding comprehension capabilities against catastrophic degradation.
- We present a stacked-isomorphic AR expansion that appends lightweight, same architecture and initialization modules to the frozen AR model, enabling generation and editing learning with the standard next-token pediction objective and no extra adapters or losses.
- During the inference phase, an implicit reasoning scheme first extracts semantic latent tokens from the frozen understanding AR and utilizes them to generate images, boosting complex-prompt alignment with zero added parameters.
- **STAR** achieves state-of-the-art performance on multimodal benchmarks (*e.g.*, GenEval **0.91**, DPG-Bench **87.44**, ImgEdit **4.34**), validating its efficacy for unified learning.

## 2 ARCHITECTURE

We introduce **STAR**, as shown in Figure 2, a novel stacked autoregressive scheme for unified multimodal learning that jointly handles visual understanding, text-to-image generation, and image editing within a single framework. Its core components comprise: (i) a vision encoder that maps images into fine-grained tokens; (ii) a stacked autoregressive model that in-place extends isomorphic layers atop a frozen pre-trained vision-language transformer, ensuring rapid convergence with minimal architecture; and (iii) a generative decoder that decodes from discrete tokens, supporting both native VQ reconstruction and diffusion-enhanced refinement for improved visual fidelity. The following subsections elaborate on the training procedure of these modules under hybrid-modality objectives.

### 2.1 VISION ENCODER

For visual input, a unified multimodal model necessitates the simultaneous incorporation of both high-level semantic information and low-level pixel details. To this end, we adopt a dual-decoupled visual representation approach to maximally preserve sufficiently fine-grained visual information for supporting downstream multimodal tasks. On one hand, since **STAR** is warm-started from a well pre-trained multimodal understanding model (Section 2.2), we directly employ the native-resolution continuous visual representations from this model for high-level semantic encoding. These features are flattened from a 2-D feature map into a 1-D token sequence, and an understanding adapter is applied to align the continuous semantic representations with the input space of the following LLM. On the other hand, for low-level pixel representations, we follow the architectural paradigm of VQ-GAN (Esser et al., 2021) and scale the original model in two aspects, proposing a more expressive vector quantizer named **STAR-VQ**. Specifically, the model size is scaled up to *1B* parameters, with the encoder comprising *0.4B* parameters and the decoder *0.6B*, while the codebook size and embedding dimension are expanded to *65,536* and *512*, respectively. After pre-training on a large-scale dataset, the $16\times$ downsampling VQ model achieves image reconstruction quality that rivals that of continuous VAEs. Using the pre-trained **STAR-VQ**, the **STAR** model tokenizes raw images into discrete codebook IDs. A generation adapter is then employed to realign the codebook embeddings corresponding to each ID to the input space of the LLM. Finally, both high-level and low-level representations are concatenated and fed into a autoregressive transformer for deep fusion.

### 2.2 STACKED AUTOREGRESSIVE MODEL

In this work, we introduce the stacked autoregressive paradigm, a principled approach that converts a pure vision–language understanding model (*e.g.*, Qwen2.5-VL (Bai et al., 2025)) into a unified architecture for comprehension and image generation by stacking additional autoregressive layers upon the base AR transformer, without introducing novel adapters, or external alignment losses. The base multimodal autoregressive transformer remains intact, as each appended layer replicates the self-attention and FFN topology, hidden dimension, and activation function of an existing layer and is initialized by copying that layer's parameters. Specifically, the parameters of the stacked autoregressive transformer are initialized from the final $N$ layers of the base autoregressive transformer, since these layers are closer to the output and therefore capture higher-level, task-relevant representations. The resulting unified model is expressed as

$$\mathcal{T}_{\text{full}} = \mathcal{T}_{\text{base}} \oplus \mathcal{T}_{\text{stack}}, \tag{1}$$

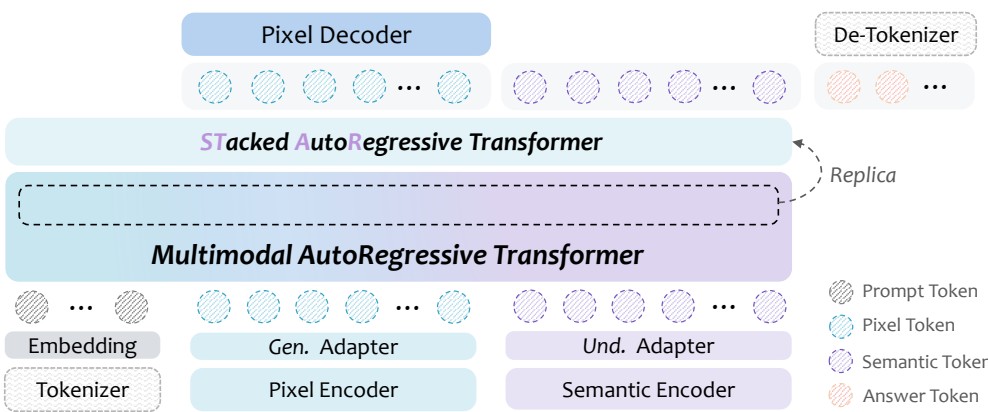

Figure 2: The overall architecture of **STAR**. The architecture integrates two visual encoder (pixel and semantic), a multimodal autoregressive transformer, a stacked autoregressive transformer, and a pixel decoder. The stacked AR is replicated from the last $N$ layer of the multimodal AR.

where $\mathcal{T}_{\text{full}}$ denotes the full autoregressive model, $\mathcal{T}_{\text{base}}$ denotes the frozen base autoregressive transformer, $\mathcal{T}_{\text{stack}}$ denotes the newly appended layers, and $\oplus$ indicates parameter-preserving concatenation along the depth dimension. Consequently, textual, visual, and cross-modal representations are mapped into a unified feature space, eliminating the 24-layer attentional adaptor used in Meta-Query and the linear projection employed in BLIP3-o and reducing connector-related parameters to exactly zero. This structural homogeneity, coupled with inherited warm-start initialisation, guarantees unimpeded gradient back-propagation and eliminates the feature discrepancy between the autoregressive token space and the continuous noise manifold characteristic of diffusion models. Optimization proceeds under a unified objective: visual inputs are quantized into the discrete token vocabulary, enabling end-to-end training with a single next-token prediction loss,

$$\mathcal{L}_{\text{NTP}} = -\sum_{t=1}^{T} \log p_{\theta}(x_t \mid x_{<t}, \mathbf{v}), \tag{2}$$

where $x_t$ denotes the target token at position $t$, $\mathbf{v}$ denotes the quantized visual tokens, and $\theta$ denotes the parameters of the stacked transformer $\mathcal{T}_{\text{full}}$. This obviates the auxiliary diffusion losses required by MetaQuery and the flow-matching objectives with their attendant task-balancing coefficients employed by BLIP3-o, yielding a succinct optimization regime with minimal hyper-parameter overhead. The unified architecture ensures parameter compactness, the consistent feature space guarantees lossless information flow, and the solitary optimisation objective delivers an efficient training regime, collectively improving overall training efficiency.

## 2.3 GENERATION DECODER

After the stacked AR transformer outputs a sequence of discrete visual tokens, it can be directly fed into the VQ decoder to decode the image. Aiming to enhance both generation quality and super-resolution capability, an additional diffusion model building upon the Lumina2-Image (Qin et al., 2025) framework is proposed to decode images from autoregressively predicted discrete tokens.

For the specific implementation of "*AR+Diffusion*", we have established systematic conclusions regarding the types of conditioning and their respective input strategies. Here, the predicted VQ tokens and the target noisy latent are strictly spatially aligned at the pixel level. For low-level tasks where pixel-wise alignment is critical, channel dimension concatenation is the common input strategy (Li et al., 2025). Specifically, let $\mathbf{z}_q \in \mathbb{R}^{K \times d}$ denote the sequence of discrete VQ tokens mapped to feature embeddings, where $K$ is the number of tokens and $d$ the embedding dimension. After reshaping and bilinear resizing we obtain a 2-D feature map $\mathbf{E}_{\text{vq}} \in \mathbb{R}^{h \times w \times d}$ that matches the spatial resolution of the noisy latent $\mathbf{x}_t \in \mathbb{R}^{h \times w \times c}$. The conditioned input to the diffusion transformer, denoted as $\mathbf{x}_{\text{in}}$, is then obtained by channel-wise concatenation:

$$\mathbf{x}_{in} = concat[\mathbf{x}_t, \ \mathbf{E}_{vq}] \in \mathbb{R}^{h \times w \times (c+d)}, \tag{3}$$

where $c$ is the original latent channel and $d$ is the codebook dimension. The model then performs super-resolution from 384 to 1024 to mitigate token explosion in AR high-resolution generation.

Figure 3: The training stages of **STAR** comprise four task-progressive phases that successively expand capability while preserving all previously acquired skills.

For image editing tasks, the source image VAE latent conditioning is designed to facilitate image consistency. In substantial modifications scenarios, such as removing or adding a large object, spatial alignment between the source image and the target noisy latent may not be strict, making sequence dimension concatenation preferable due to its flexible control over channel concatenation (Huang et al., 2025b; Guo & Lin, 2024). Thus, we choose to concatenate the VAE latent of the source image with the noisy latent along the sequence dimension. Since there is no source image in text-to-image generation, we introduce a zero latent as an unconditional placeholder, allowing joint training of both image editing and text-to-image tasks with a shared diffusion decoder. Consequently, our diffusion decoder accommodates three types of conditioning: text, resized VQ embeddings, and source image VAE latent. Our experimental results consistently validate our theoretical analysis and systematic findings about conditioning strategies.

# 3 TRAINING AND INFERENCE RECIPE

## 3.1 TRAINING RECIPE

To achieve multi-stage, incremental enhancement of multi-task capabilities, we adopt a four-stages progressive training framework, whose workflow is illustrated in Figure 3.

**Stage 1: Pixel-level Vector Quantization Pretraining.** The objective of this stage is to train a vector quantization model from scratch to achieve a higher-fidelity discrete representation of low-level information of raw images. As introduced in Section 2.1, **STAR-VQ** is designed to reduce quantization information loss by scaling up both model parameter size and codebook dimension. However, such expansion often leads to increased training difficulty and decreased codebook utilization, *i.e.*, codebook collapse. To address this, we draw inspiration from (Chang et al., 2025) and employ an additional codebook projector (2 DiT-blocks) during training, which compresses and reconstructs the codebook, and then performes image reconstruction training based on the reconstructed codebook. This stage involves training on a combined corpus of ImageNet (Deng et al., 2009) and OpenImages (Kuznetsova et al., 2020) for 120 epochs.

**Stage 2: Stacked AR Text-to-Image Pretraining.** To endow the frozen multimodal backbone with text-to-image generation, we stack isomorphically designed AR layers and train them exclusively on 60M general plus 0.6M high-quality synthetic image–text pairs. The pretrained **STAR-VQ** quantises images into discrete tokens; text and visual tokens are fed to both the base and the stacked AR modules, and next-token cross-entropy loss updates only the newly added parameters, preventing any semantic drift of the original understanding layers.

**Stage 3: AR-Diffusion Alignment Training.** In this stage, only the diffusion decoder is pretrained for decoding VQ embeddings, with all other modules frozen and the VQ decoder replaced by the diffusion decoder. Images with a total pixel count close to $512 \times 512$ are used, and the training data comprises a 10M-image subset from the text-to-image dataset.

**Stage 4: Unified Text-to-Image and Edit Instruction Tuning.** In this stage, the diffusion decoder and Stacked-AR are jointly trained on both generation and editing data, aiming to let Stacked-AR impart image editing capabilities while maintaining its text-to-image generation performance. To prevent interference from the diffusion decoder's loss on Stacked-AR training, a stop-gradient

Table 1: Evaluation on multimodal understanding benchmarks.

| Model | #LLM | MMB | MMStar | MathVista | SEED | MME-P | MMMU | OCRBench | POPE | DocVQA |
|---|---|---|---|---|---|---|---|---|---|---|
| Seed-X (Ge et al., 2024) | 13B | 70.1 | - | - | 66.5 | 1457.0 | 35.6 | - | - | - |
| EMU3 (Wang et al., 2024) | 8B | 58.5 | - | - | 68.2 | 1243.8 | 31.6 | 68.7 | 85.2 | - |
| MetaMorph (Tong et al., 2024) | 8B | 75.2 | - | - | 71.8 | - | 41.8 | - | - | - |
| Janus (Wu et al., 2024) | 1.3B | 75.5 | - | - | 63.7 | 1338.0 | 30.5 | - | 87.0 | - |
| Janus-Pro (Chen et al., 2025c) | 7B | 79.2 | 87.4 | - | 72.1 | 1567.1 | 41.0 | - | - | - |
| BLIP3-o (Chen et al., 2025a) | 8B | 83.5 | - | - | 77.5 | 1682.6 | 50.6 | - | - | - |
| Show-o2 (Xie et al., 2025) | 7B | 79.3 | 56.6 | - | 69.8 | 1620.0 | 48.9 | - | - | - |
| MetaQuery-XL (Pan et al., 2025) | 7B | 83.5 | - | - | 76.9 | 1685.2 | 58.6 | - | - | - |
| Bagel (Deng et al., 2025) | 14B | 85.0 | - | 73.1 | - | 1687.0 | 55.3 | - | - | - |
| Ovis-U1 (Wang et al., 2025) | 1.5B | 77.8 | - | 69.4 | - | - | 51.1 | 88.3 | - | - |
| ILLUME+ (Huang et al., 2025a) | 3B | 80.8 | - | - | 73.3 | 1414.0 | 44.3 | 67.2 | 87.6 | 80.8 |
| X-Omni (Geng et al., 2025) | 7B | 74.8 | - | - | 74.1 | - | - | 70.4 | 89.3 | 88.6 |
| *STAR-3B* | 3B | 80.1 | 55.8 | 62.3 | 74.0 | 1592.3 | 53.1 | 79.7 | 85.9 | 93.9 |
| *STAR-7B* | 7B | 83.9 | 63.9 | 68.1 | 77.0 | 1690.1 | 58.6 | 86.4 | 86.6 | 95.7 |

operation is applied when Stacked-AR outputs VQ embeddings. The training data consists of a 6M-image subset from the text-to-image dataset and a 4M-image subset from the editing dataset.

## 3.2 INFERENCE RECIPE

Upon completion of training, we evaluate the model on three families of tasks: multimodal understanding, text-to-image generation, and image editing, within a single forward pipeline. For understanding, the frozen base autoregressive transformer receives an image-text question and directly emits the textual answer. For generation, the text prompt is processed sequentially by the base and the stacked AR layers, which progressively predict the discrete image-token sequence. The resulting tokens are fed to the generative decoder to reconstruct a image. For editing, the original image and the textual instruction are concatenated and processed by the stacked AR model, yielding an edited token sequence that is decoded into a semantically consistent result. When the prompt demands external knowledge or complex reasoning, we invoke an *implicit-token-reasoning* mechanism: the base AR first infers an intermediate latent-token sequence that encodes the required knowledge, and this sequence is supplied as an conditioning signal to the stacked AR for image generation. Experiments show that this strategy yields substantial gains on generation benchmarks that probe world knowledge and compositional semantics.

## 4 EXPERIMENT

### 4.1 DATA COMPOSITION

**Text-to-Image Generation Data.** This dataset is primarily used for training text-to-image generation tasks. We collected publicly available text-image pairs and leveraged powerful generative models such as FLUX (Labs, 2024), GPT-4o (OpenAI, 2025), and Midjourney. Ultimately, we constructed a total of 60M text-to-image generation data.

**Image Edit Data.** In our experiments, we utilize a diverse set of pre-trained image editing datasets to support both the Stage-3 and Stage-4 training. The publicly available image editing data comprises main sources from UltraEdit (Zhao et al., 2024), HQ-Edit (Hui et al., 2024), and Omni-Edit (Wei et al., 2024), approximately 4M samples are selected. Also, we re-synthesized ground truth images using the GPT-4o (OpenAI, 2025) on approximately 300K proprietary samples. This combination of large-scale public and private datasets, along with high-fidelity ground truth synthesis, ensures robust and comprehensive supervision for image editing task throughout training.

### 4.2 EVALUATION SETUP

**Image Understanding Evaluation.** We assess image-understanding capabilities on the nine standardized benchmarks: MMBench-EN(Liu et al., 2023), MMStar(Chen et al., 2024), MathVista (Lu et al., 2023), SEEDBench (Li et al., 2023), MME (Fu et al., 2023), MMMU (Yue et al., 2024), OCRBench (Liu et al., 2024), POPE (Yifan et al., 2023), and DocVQA (Mathew et al., 2021).

Table 2: Comparison with state-of-the-art text-to-image generation methods on GenEval (Ghosh et al., 2023) and DPG-Bench (Hu et al., 2024).

| Method | GenEval | | | | | | | DPG-Bench | | | | | |
|---|---|---|---|---|---|---|---|---|---|---|---|---|---|
| | Single | Two | Count. | Colors | Pos. | Color Attr. | Overall | Global | Entity | Attr. | Relation | Other | Overall |
| *Gen. Only Models* | | | | | | | | | | | | | |
| SDXL (Podell et al., 2024) | 0.98 | 0.74 | 0.39 | 0.85 | 0.15 | 0.23 | 0.55 | 83.27 | 82.43 | 80.91 | 86.76 | 80.41 | 74.65 |
| DALL-E (OpenAI, 2024b) | 0.96 | 0.87 | 0.47 | 0.83 | 0.43 | 0.45 | 0.67 | 90.97 | 89.61 | 88.39 | 90.58 | 89.83 | 83.50 |
| SD3-medium (Esser et al., 2024) | 0.99 | 0.94 | 0.72 | 0.89 | 0.33 | 0.60 | 0.74 | 87.90 | 91.01 | 88.83 | 80.70 | 88.68 | 84.08 |
| FLUX.1-dev (Labs, 2024) | 0.98 | 0.93 | 0.75 | 0.93 | 0.68 | 0.65 | 0.82 | 82.10 | 89.50 | 88.70 | 91.10 | 89.40 | 84.00 |
| OmniGen2 (Wu et al., 2025) | 0.99 | 0.96 | 0.74 | 0.98 | 0.72 | 0.75 | 0.86 | 88.81 | 88.83 | 90.18 | 89.37 | 90.27 | 83.57 |
| *Unified Models* | | | | | | | | | | | | | |
| Emu3 (Wang et al., 2024) | 0.99 | 0.81 | 0.42 | 0.80 | 0.49 | 0.45 | 0.66 | 85.21 | 86.68 | 86.84 | 90.22 | 83.15 | 80.60 |
| ILLUME+ (Huang et al., 2025a) | 0.99 | 0.88 | 0.62 | 0.84 | 0.42 | 0.53 | 0.72 | - | - | - | - | - | - |
| Janus-Pro (Chen et al., 2025c) | 0.99 | 0.89 | 0.59 | 0.90 | 0.79 | 0.66 | 0.80 | 86.90 | 88.90 | 89.40 | 89.32 | 89.48 | 84.19 |
| MetaQuery (Pan et al., 2025) | - | - | - | - | - | - | 0.80 | - | - | - | - | - | 82.05 |
| BLIP3-o (Chen et al., 2025a) | - | - | - | - | - | - | 0.84 | - | - | - | - | - | 81.60 |
| UniWorld-V1 (Lin et al., 2025) | 0.99 | 0.93 | 0.81 | 0.89 | 0.74 | 0.71 | 0.84 | 83.64 | 88.39 | 88.44 | 89.27 | 87.22 | 81.38 |
| Mogao (Liao et al., 2025) | 1.00 | 0.97 | 0.83 | 0.93 | 0.84 | 0.80 | 0.89 | 82.37 | 90.03 | 88.26 | 93.18 | 85.40 | 84.33 |
| BAGEL (Deng et al., 2025) | 0.98 | 0.95 | 0.84 | 0.95 | 0.78 | 0.77 | 0.88 | 88.94 | 90.37 | 91.29 | 90.82 | 88.67 | 85.07 |
| Show-o2 (Xie et al., 2025) | 1.00 | 0.87 | 0.58 | 0.92 | 0.52 | 0.62 | 0.76 | 89.00 | 91.78 | 89.96 | 91.81 | 91.64 | 86.14 |
| GPT-4o (OpenAI, 2025) | 0.99 | 0.92 | 0.85 | 0.92 | 0.75 | 0.61 | 0.84 | 82.27 | 91.27 | 87.67 | 93.85 | 88.71 | 86.23 |
| X-Omni (Geng et al., 2025) | 0.98 | 0.95 | 0.75 | 0.91 | 0.71 | 0.68 | 0.83 | 84.80 | 92.59 | 90.63 | 94.75 | 84.20 | **87.65** |
| Ovis-U1 (Wang et al., 2025) | 0.98 | 0.98 | 0.90 | 0.92 | 0.79 | 0.75 | 0.89 | 82.37 | 90.08 | 88.68 | 93.35 | 85.20 | 83.72 |
| *STAR-3B* | 0.98 | 0.87 | 0.85 | 0.91 | 0.79 | 0.76 | 0.86 | 93.00 | 90.49 | 91.71 | 90.72 | 92.75 | 87.30 |
| *STAR-7B* | 0.98 | 0.94 | 0.90 | 0.92 | 0.91 | 0.80 | **0.91** | 94.97 | 92.91 | 91.62 | 94.30 | 83.82 | 87.44 |

Table 3: Comparison of world knowledge reasoning on WISE (Niu et al., 2025).

| Methods | Cultural | Time | Space | Biology | Physics | Chemistry | Overall |
|---|---|---|---|---|---|---|---|
| *Gen. Only Models* | | | | | | | |
| SD-XL (Podell et al., 2024) | 0.43 | 0.48 | 0.47 | 0.44 | 0.45 | 0.27 | 0.43 |
| SD-3.5-large (Esser et al., 2024) | 0.44 | 0.50 | 0.58 | 0.44 | 0.52 | 0.31 | 0.46 |
| FLUX.1-dev (Labs, 2024) | 0.48 | 0.58 | 0.62 | 0.42 | 0.51 | 0.35 | 0.50 |
| *Unified Models* | | | | | | | |
| Emu3 (Wang et al., 2024) | 0.34 | 0.45 | 0.48 | 0.41 | 0.45 | 0.27 | 0.39 |
| Janus-Pro-7B (Chen et al., 2025c) | 0.30 | 0.37 | 0.49 | 0.36 | 0.42 | 0.26 | 0.35 |
| MetaQuery-XL (Pan et al., 2025) | 0.56 | 0.55 | 0.62 | 0.49 | 0.63 | 0.41 | 0.55 |
| BLIP3-o (Chen et al., 2025a) | - | - | - | - | - | - | 0.62 |
| BAGEL (Deng et al., 2025) | 0.76 | 0.69 | 0.75 | 0.65 | 0.75 | 0.58 | 0.70 |
| GPT-4o (OpenAI, 2025) | 0.94 | 0.64 | 0.98 | 0.93 | 0.98 | 0.95 | **0.89** |
| *STAR-3B* | 0.58 | 0.54 | 0.48 | 0.49 | 0.51 | 0.54 | 0.52 |
| *STAR-7B* | 0.61 | 0.67 | 0.61 | 0.74 | 0.69 | 0.66 | 0.66 |

**Text-to-image Evaluation.** This task evaluates semantic consistency on GenEval (Ghosh et al., 2023) (553 prompts) and DPG-Bench (Hu et al., 2024) (1065 prompts), and world knowledge is measured on WISEBench (Niu et al., 2025) (1000 prompts).

**Image Editing Evaluation.** Image-editing capability is assessed on MagicBrush (Zhang et al., 2023) (1,000 pairs) and ImgEdit (Ye et al., 2025) (737 pairs), the latter covering object-level, background, style, action, and composite manipulations. For MagicBrush we report CLIP-I, DINO (content preservation), and L1 (pixel-level fidelity).

### 4.3 MAIN RESULTS

*Image Understanding.* Thanks to our task-progressive training regime, the proposed model family can be grafted onto any state-of-the-art multimodal understanding backbone without impairing its original capability. By freezing the comprehension parameters and augmenting capacity through stacked autoregressive modules, we retain the full representational strength of the upstream encoder while equipping it with high-fidelity generation. Consequently, our checkpoints inherit both the semantic richness of the underlying understanding network (Bai et al., 2025) and the generative power of contemporary SOTA architectures. As shown in Table 1, they achieve competitive or leading results on a broad range of understanding benchmarks, including MMStar, SEED, MME and OCRBench, demonstrating that task-progressive extension yields a unified system that excels simultaneously in comprehension and generation.

*Image Generation.* We comprehensively evaluated the generative capability of our model on three public benchmarks: GenEval and DPG-Bench for prompt–image alignment, and WISE for world-

Table 4: Comparison of image editing performance on the MagicBrush (Zhang et al., 2023).

| | MagicBrush | Instruct-Pix2Pix | UltraEdit | ICEdit | OmniGen | UniReal | BAGEL | *STAR-3B* | *STAR-7B* |
|---|---|---|---|---|---|---|---|---|---|
| L1 ↓ | 0.074 | 0.114 | 0.066 | 0.060 | 0.116 | 0.081 | 0.074 | **0.056** | **0.060** |
| CLIP-I ↑ | 0.908 | 0.851 | 0.904 | 0.928 | 0.863 | 0.903 | 0.914 | **0.934** | **0.931** |
| DINO ↑ | 0.847 | 0.744 | 0.852 | 0.853 | 0.821 | 0.837 | 0.827 | **0.857** | **0.853** |

Table 5: Comparison of image editing performance on ImgEdit-Bench (Ye et al., 2025).

| Model | Add | Adjust | Extract | Replace | Remove | Background | Style | Hybrid | Action | Overall |
|---|---|---|---|---|---|---|---|---|---|---|
| *Edit. Only Models* | | | | | | | | | | |
| MagicBrush (Zhang et al., 2023) | 2.84 | 1.58 | 1.51 | 1.97 | 1.58 | 1.75 | 2.38 | 1.62 | 1.22 | 1.90 |
| Instruct-Pix2Pix (Brooks et al., 2023) | 2.45 | 1.83 | 1.44 | 2.01 | 1.50 | 1.44 | 3.55 | 1.20 | 1.46 | 1.88 |
| AnyEdit (Yu et al., 2025) | 3.18 | 2.95 | 1.88 | 2.47 | 2.23 | 2.24 | 2.85 | 1.56 | 2.65 | 2.45 |
| UltraEdit (Zhao et al., 2024) | 3.44 | 2.81 | 2.13 | 2.96 | 1.45 | 2.83 | 3.76 | 1.91 | 2.98 | 2.70 |
| Step1X-Edit (Liu et al., 2025) | 3.88 | 3.14 | 1.76 | 3.40 | 2.41 | 3.16 | 4.63 | 2.64 | 2.52 | 3.06 |
| ICEdit (Zhang et al., 2025) | 3.58 | 3.39 | 1.73 | 3.15 | 2.93 | 3.08 | 3.84 | 2.04 | 3.68 | 3.05 |
| *Unified Models* | | | | | | | | | | |
| GPT-4o (OpenAI, 2025) | 4.61 | 4.33 | 2.90 | 4.35 | 3.66 | 4.57 | 4.93 | 3.96 | 4.89 | 4.20 |
| OmniGen (Xiao et al., 2024) | 3.47 | 3.04 | 1.71 | 2.94 | 2.43 | 3.21 | 4.19 | 2.24 | 3.38 | 2.96 |
| BAGEL (Deng et al., 2025) | 3.56 | 3.31 | 1.70 | 3.30 | 2.62 | 3.24 | 4.49 | 2.38 | 4.17 | 3.20 |
| UniWorld-V1 (Lin et al., 2025) | 3.82 | 3.64 | 2.27 | 3.47 | 3.24 | 2.99 | 4.21 | 2.96 | 2.74 | 3.26 |
| OmniGen2 (Wu et al., 2025) | 3.57 | 3.06 | 1.77 | 3.74 | 3.20 | 3.57 | 4.81 | 2.52 | 4.68 | 3.44 |
| Ovis-U1 (Wang et al., 2025) | 4.13 | 3.62 | 2.98 | 4.45 | 4.06 | 4.22 | 4.69 | 3.45 | 4.61 | 4.00 |
| *STAR-3B* | 4.26 | 4.06 | 3.78 | 4.46 | 4.34 | 4.19 | 4.53 | 3.29 | 4.38 | 4.14 |
| *STAR-7B* | 4.33 | 4.19 | 4.19 | 4.59 | 4.58 | 4.36 | 4.59 | 3.67 | 4.60 | **4.34** |

Figure 4: (a) Qualitative comparison results. The proposed diffusion decoder yields sharper textures and finer details than the VQ decoder, demonstrating its superior high-fidelity generation capability. (b) Reasoning mode can utilize the world knowledge of MLLM for reasoning-based text-to-image.

knowledge reasoning. For the latter, we further activated the proposed implicit-reasoning pipeline at inference to mitigate visual–semantic distributional shifts. As reported in Table 2, the model establishes a new state-of-the-art on GenEval with 0.91 (2.0% over the prior best Ovis-U1), while delivering competitive scores on DPG-Bench in Table 2. As shown in the Table 3, implicit inference reasoning on WISE (Niu et al., 2025) attains a score of 0.66, confirming that latent-token mediation significantly enhances compositional and knowledge-intensive generation as shown in Figure 4.

***Image Editing.*** Tables 4 and 5 present the evaluation results for image editing capabilities on MagicBrush and ImgEdit, respectively. On ImgEdit, we compare our model with existing unified models. For the MagicBrush, in addition to unified models, we also include comparisons with specialized image editing models such as Instruct-Pix2Pix, UltraEdit (Zhao et al., 2024), and ICEdit. The performance of previous models on ImgEdit is referenced from Ovis-u1, while the MagicBrush results are computed by us. Overall, our model achieves strong performance across both benchmarks.

## 4.4 ABLATION STUDIES

***Different type of VQ tokenizer.*** To obtain higher-fidelity discrete image representations, we replaced the conventional VQGAN reconstruction tokenizer with the ***STAR-VQ***. Table 6a compares the two approaches under a controlled 3B architecture trained on 6M synthetic images and evaluated on GenEval. ***STAR-VQ*** raises the GenEval score from 0.414 to 0.439, confirming that its larger and higher-dimensional codebook yields finer-grained visual tokens. The gain indicates that

Table 6: Ablation studies of VQ and stacked AR.

(a) Different type of VQ tokenizer. Size and Dim represent the codebook size and dimension of token.

| VQ Type | Size | Dim | GenEval |
|---------|------|-----|---------|
| VQGAN | 16384 | 8 | 0.414 |
| *STAR-VQ* | 65536 | 512 | **0.439** |

(b) The number of layers.

| Layer | GenEval |
|-------|---------|
| 8 | 0.347 |
| **16** | **0.439** |
| 32 | 0.410 |
| 36 | 0.394 |

(c) Ablation of initial of stacked AR.

| Init From | GenEval |
|-----------|---------|
| Rand | 0.374 |
| LLM | 0.403 |
| **VLM** | **0.439** |

Table 7: Ablation studies of diffusion decoder. All results are obtained after stage 3 or 4 training.

(a) The impact of the diffusion decoder.

| Stage | Decoder Type | Size | GenEval |
|-------|--------------|------|---------|
| Stage 3 | VQ Dec. | 384 | 0.723 |
| Stage 3 | Diffusion | 1024 | **0.756** |
| Stage 4 | VQ Dec. | 384 | 0.858 |
| Stage 4 | Diffusion | 1024 | **0.868** |

(b) The input way from AR to DiT.

| Input of VQ emb | GenEval |
|-----------------|---------|
| Text-wise Concat | 0.703 |
| Sequence-wise Concat | 0.712 |
| **Channel-wise Concat** | **0.756** |

the augmented discrete vocabulary supplies the autoregressive generator with more precise spatial and semantic cues, ultimately translating into superior synthesis quality.

***The number of layers in the stacked AR.*** To identify the optimal depth of the stacked autoregressive transformer, we perform layer-wise ablation on the ***STAR-3B*** model trained with 6M data and evaluated on GenEval. As reported in Table 6b, accuracy increases with depth until 16 layers, which attains the highest score of 0.439, and declines thereafter. This inverted-U profile indicates that shallow stacks lack the capacity to model the target distribution, whereas deeper model suffer from diminishing gradient signals that progressively weaken updates and ultimately degrade performance.

***The initialization strategy of stacked AR.***

To determine the optimal initialization for the stacked autoregressive modules, we train ***STAR-3B*** model on 6 M generation data and evaluate on GenEval. As shown in Table 6c, VLM-based initialization reaches 0.439, exceeding LLM-based initialization (0.403) and random initialization (0.374), respectively. Initializing stacked AR layers with parameters homologous to the primary AR leverages strong inherent feature-space alignment, thereby accelerating convergence and enhancing generation quality by eliminating re-alignment and directly exploiting learned representational priors.

***The impact of Diffusion decoder.*** To elucidate the role of the diffusion decoder in autoregressive text-to-image generation, we replace the vanilla VQ decoder with a diffusion decoder. As reported in Table 7a, the switch yields consistent gains on GenEval (0.03 at stage3 and 0.01 at stage4), corroborating that iterative denoising recovers high-frequency information lost during quantization. Qualitative visualizations in Figure 4 further reveal markedly sharper textures, cleaner edges and suppressed aliasing artifacts, validating that the diffusion translates coarse AR tokens into photorealistic outputs with enhanced pixel fidelity.

***The input strategy of Diffusion decoder.*** We ablate three strategies for feeding AR tokens into the diffusion decoder: (i) text-wise concatenation; (ii) sequence-wise concatenation; and (iii) channel-wise concatenation after resizing. They are trained on a subset of the dataset. Table 7b shows that strategy channel-wise concatenation after resizing achieves the highest GenEval score (0.756), establishing it as the preferred interface.

# 5 CONCLUSION

In this work, we present ***STAR***, a task-progressive framework that unifies multimodal understanding, generation, and editing within a single MLLM without sacrificing any capability. By freezing the original autoregressive model and incrementally stacking isomorphic AR layers, ***STAR*** eliminates cross-task gradient interference. We further equip the generative AR with ***STAR-VQ***, a high-capacity tokenizer that boosts discrete-image fidelity, and an implicit inference mechanism that leverages intermediate semantic tokens to handle complex prompts. Extensive experiments show that ***STAR*** sets new state-of-the-art results on both comprehension and generation tasks. These findings validate that orderly, interference-free expansion is a viable route toward scalable and sustainable general-purpose multimodal systems.

## 6 ETHICS STATEMENT

This study strictly follows the ICLR Code of Ethics. No human-subject or animal experimentation was conducted. All datasets were obtained and used in accordance with their respective licenses and privacy policies. We implemented measures to prevent discriminatory bias and did not collect or process any personally identifiable information. No experimental procedures posed privacy or security risks. Transparency and research integrity were maintained throughout the project.

## 7 REPRODUCIBILITY STATEMENT

The main paper describes the detailed design and training process of our method. The appendix further provides detailed experimental hyperparameter settings, providing readers with all the information necessary to reproduce the reported results. To ensure full reproducibility, we will release the full source code, trained models, and configuration files immediately after review, so that the community can reproduce our experiments and fully verify our findings.

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

## A    RELATED WORK

**Training Unified Multimodal Models from Scratch.**    Recent efforts toward unified multimodal models typically begin with a pre-trained large language model (LLM) and fine-tune it on paired understanding and image-generation objectives. SEED-X (Ge et al., 2024), Emu (Sun et al., 2023), and MetaMorph (Tong et al., 2024) regress continuous image features. Chameleon (Team, 2024), EMU3 (Wang et al., 2024), and the Janus family (Wu et al., 2024; Chen et al., 2025c) encode images into discrete tokens and unify image and text token prediction under a single next-token prediction objective. DreamLLM (Dong et al., 2023), Show-o (Xie et al., 2024), Show-o2 (Xie et al., 2025), and Transfusion (Zhou et al., 2024) further combine diffusion and next-token losses within one framework. An alternative line appends an external diffusion model after the LLM, like Ovis-U1 (Wang et al., 2025), requires training an intermediate adapter. BAGEL (Deng et al., 2025) and Mogao (Liao et al., 2025) introduce MoE or MoT routers to decouple parameters so that distinct experts handle distinct tasks, while X-Omni (Geng et al., 2025) adds a reinforcement-learning stage to boost generation quality. Despite their effectiveness, these approaches force the backbone to master multiple generation targets, complicating multi-task balancing and motivating our task-progressive alternative.

**Training Unified Multimodal Models via Warm-Start Adaptation.**    An alternative line of research freezes the large multimodal backbone and grafts on lightweight generative modules. LM-Fusion (Shi et al., 2024) trains parallel FFN/QKV experts that share the frozen LLM topology, yet every new backbone necessitates a complete set of newly trained generative parameters, raising computational cost. MetaQuery (Pan et al., 2025) prepends learnable queries to the fixed MLLM and feeds their outputs into a connector that drives a DiT generative model, whereas BLIP3-o (Chen et al., 2025a) directly conditions a diffusion model on MLLM features and supervises the diffusion output with a flow-matching loss against CLIP (Radford et al., 2021) image embeddings. Both approaches require a feature converter to map autoregressive outputs into the diffusion latent space and introduce auxiliary objectives, *e.g.*, diffusion or flow-matching losses, that create optimization paths diverging from the original next-token prediction objective. These shortcomings motivate our stacked-autoregressive task-progressive paradigm, which expands generation capacity while preserving the fundamental comprehension ability.

## B    OVERALL ARCHITECTURAL PARAMETERS

Overall architectural parameters are summarised in Table 8, confirming the compactness of the proposed design. **STAR-3B** extends the Qwen2.5-VL-3B (Bai et al., 2025) vision–language model by appending a Stacked AR module that replicates the final 16 layers of the VLM for initialization, contributing 1.5 B additional parameters. **STAR-7B**, built upon Qwen2.5-VL-7B (Bai et al., 2025), mirrors the last 14 layers of the VLM, adding 3 B parameters. Both variants share an identical VQ tokenizer and diffusion decoder.

Table 8: Overall architecture constituents and parameter counts.

| Model | VLM | Pixel-Enc. | Gen-Adapter | Stacked-AR | VQ-Dec. | Diff-Dec. |
|---|---|---|---|---|---|---|
| *STAR-3B* | Qwen2.5-VL-3B | 0.4B | 5M | 1.2B (16 Layer) | 0.6B | 2.6B |
| *STAR-7B* | Qwen2.5-VL-7B | 0.4B | 38M | 3B (14 Layer) | 0.6B | 2.6B |

## C    TRAINING STRATEGIES AT EACH STAGE

We also list the training strategies we used in each stage in Table 9.

## D    ABLATION OF REASONING

We have incorporated ablation experiments for the reasoning mechanism on **STAR-7B**. On the WISE evaluation set, we compared performance before and after adding the reasoning mechanism. The experimental results are shown in the Table 10. As can be seen from the table, adding the reasoning

Table 9: Training strategies at each stage.

| Hyper-Parameter | Stage 1 | Stage 2 | Stage 3 | Stage 4 |
|---|---|---|---|---|
| Learning Rate | 1e-4 | 1e-3 | 1e-4 | 1e-3 |
| LR Scheduler | cosine | constant | constant | constant |
| Optimizer | AdamW | Adamw | AdamW | AdamW |
| Batch Size | 256 | 4096 | 2048 | 4608 |
| Training Steps | 1406K | 20K | 4K | 8K |
| VQ Image *Res.* | 256×256 | 384×384 | 384×384 | 384×384 |
| Diffusion *Res.* | / | / | 512×512 | 1024×1024 |

Table 10: Ablation of reasoning strategy on WISE.

| Method | Cultural | Time | Space | Biology | Physics | Chemistry | Overall |
|---|---|---|---|---|---|---|---|
| w/o Reasoning | 0.49 | 0.52 | 0.45 | 0.48 | 0.51 | 0.35 | 0.46 |
| w/ Reasoning | 0.61 | 0.67 | 0.61 | 0.74 | 0.69 | 0.66 | 0.66 |

mechanism yields noticeable improvements across every subtask in the WISE benchmark, ultimately achieving a 0.2 increase in the overall metric. This improvement stems from our approach leveraging the foundational reasoning capabilities of the VLM, which is extensively trained on broad world knowledge. Consequently, when processing abstract textual prompts, such as "The fastest land animal.", the model first employs VLM to inference the specific target subject "Cheetah". Following this, the generation model produces images that most closely match the prompt.

## E  ABLATION OF MULTI-STAGE SEPARATE TRAINING

We compared single-stage joint training with multi-stage separate training paradigms to demonstrate the advantages of the multi-stage approach. Single-stage joint training involves simultaneously training the stacked-AR module and diffusion decoder starting from the pre-training phase. While multi-stage separate training refers to our main approach, where only the stacked-AR module is trained during the pre-training phase, followed by training the diffusion decoder in the subsequent stage. The experimental results are shown in the Table 11. As seen, on the same 3B base model, the performance of single-stage joint training is significantly lower than that of multi-stage separate training, with differences of 0.07 and 5.27 on GenEval and DPG-Bench, respectively. Our analysis indicates that during the early stages of single-stage training, the stacked-AR module possesses minimal ability to model images autoregressively. Consequently, its predicted token representations become chaotic, negatively impacting the diffusion decoder's inherent capabilities. This further demonstrates that multi-stage separate training minimizes interference between module training, ultimately yielding superior overall generation performance.

Table 11: Ablation of multi-stage separate training.

| Method | Model size | GenEval | DPG-Bench |
|---|---|---|---|
| Single-stage joint training | 3B | 0.79 | 82.03 |
| Multi-stage separate training | 3B | 0.86 | 87.30 |

## F  LLM USAGE

We utilized Large Language Models (LLMs) to assist in language polishing and readability enhancement of the manuscript. The LLM contributed to tasks such as sentence rephrasing, grammar correction, and improving textual flow, without involvement in research ideation, methodology, or experimental design. All scientific content, analyses, and interpretations were exclusively developed by the authors. We take full responsibility for the final content and confirm that LLM-assisted text complies with ethical standards and does not introduce plagiarism or scientific misconduct.

## G  MORE QUALITATIVE RESULTS

We give more qualitative results on text-to-image generation (Figure 5 and 6) and image-editing (Figure 7). We also presented some examples of failure in the Figure 8.

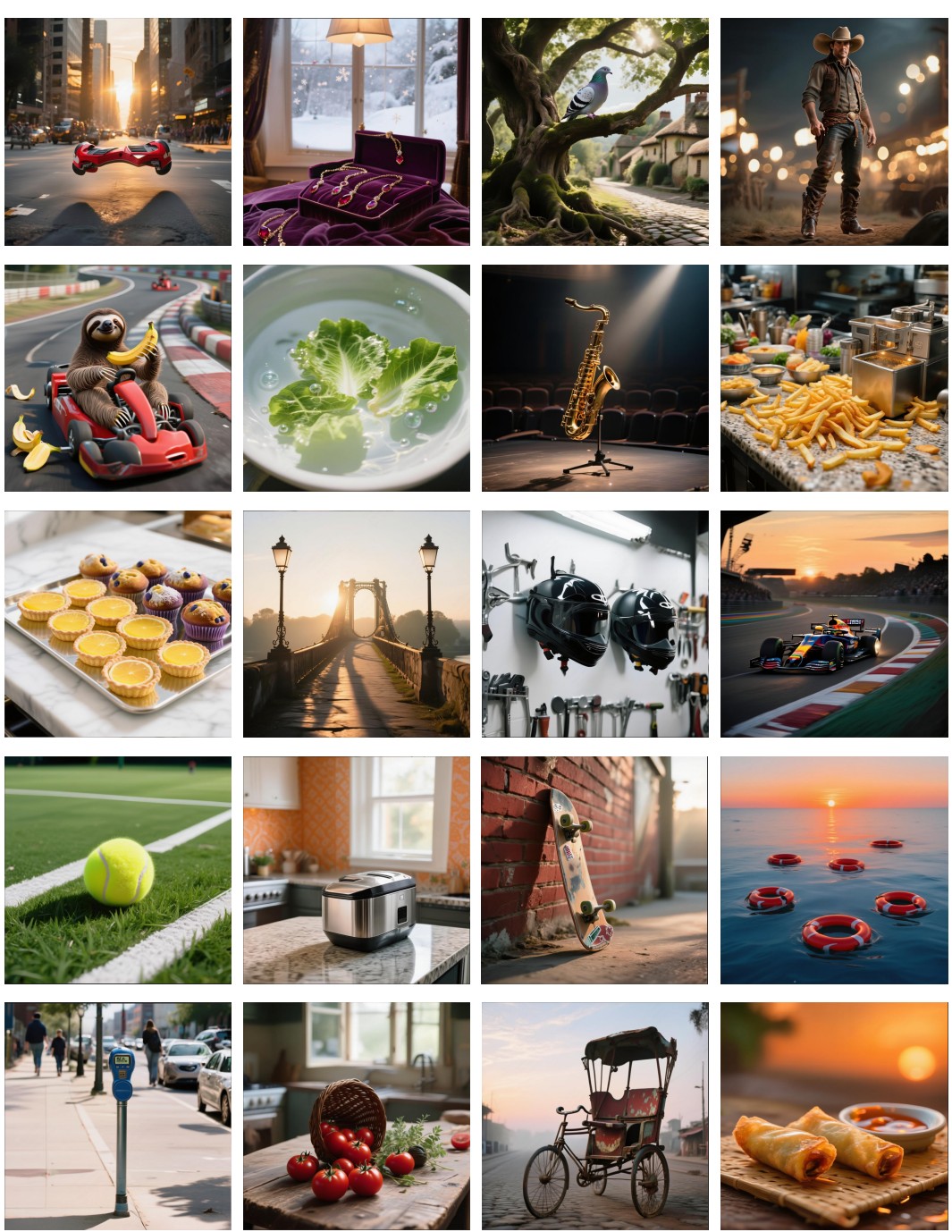

Figure 5: More qualitative results on text-to-image generation

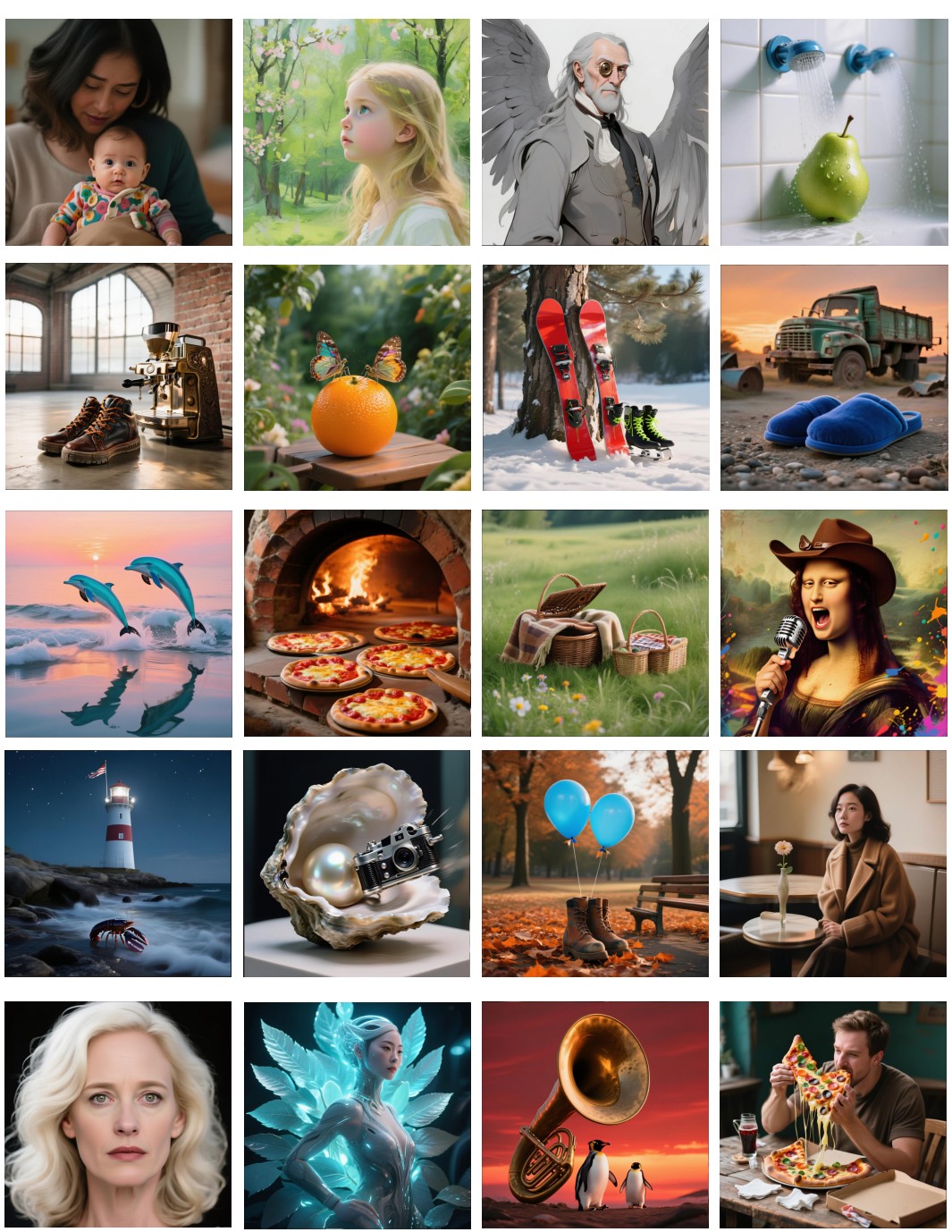

Figure 6: More qualitative results on text-to-image generation

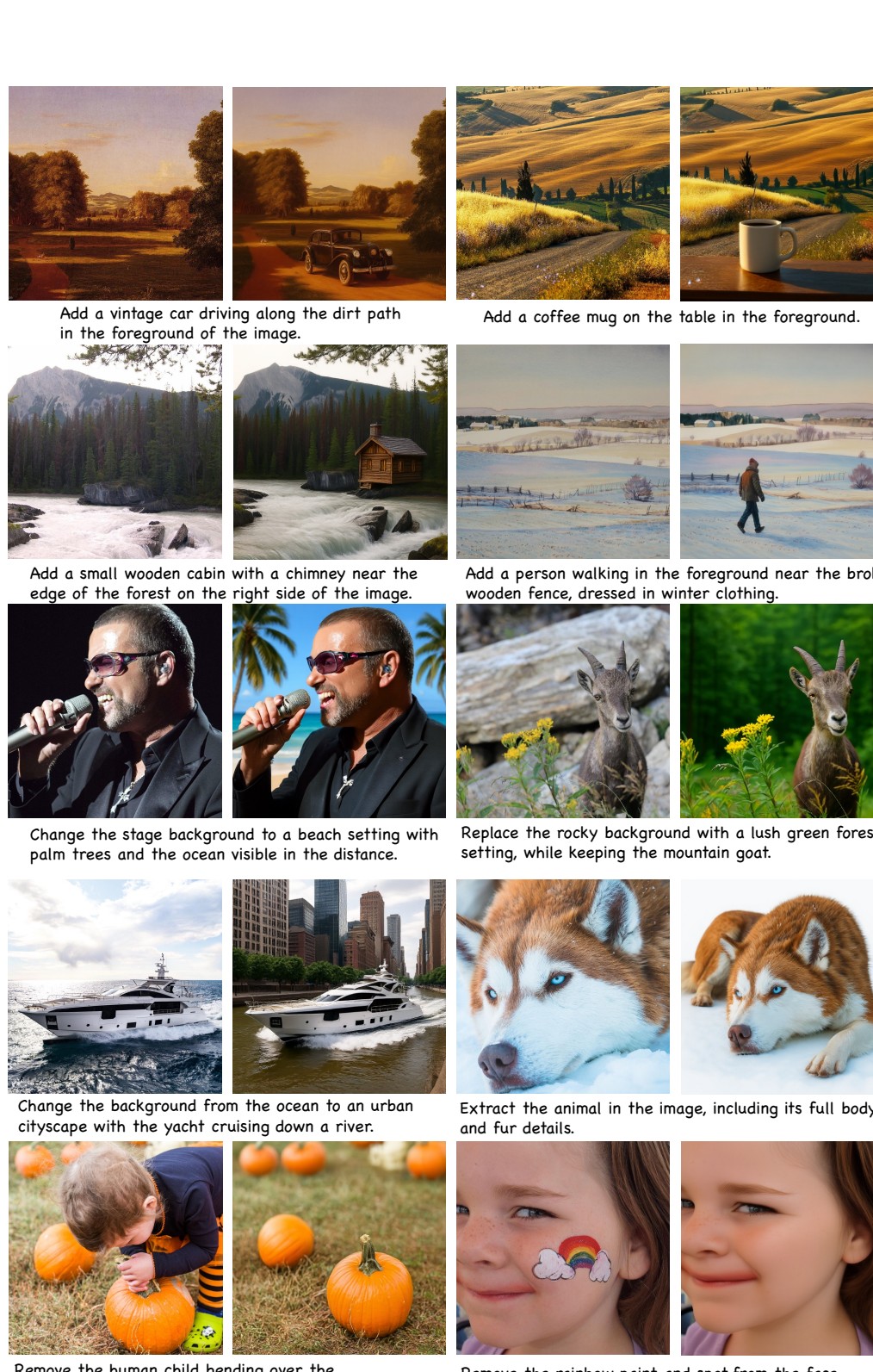

Figure 7: More qualitative results on image editing

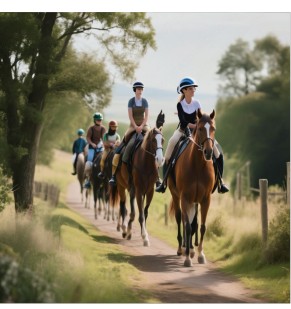

A scenic trail where a group of riders are mounted on their horses ...

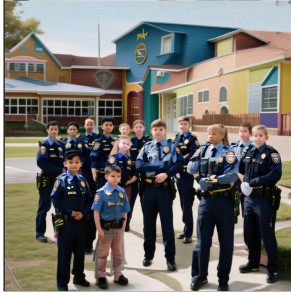

Ten School Resource Officers are set to serve in seven local school districts, as part of the School Resource Officer (SRO) Program ...

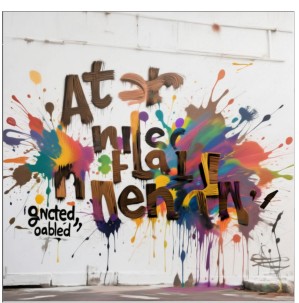

On a stark white wall, the phrase "Art is never finished, only abandoned" comes to life through an array of dynamic paint splatters ...

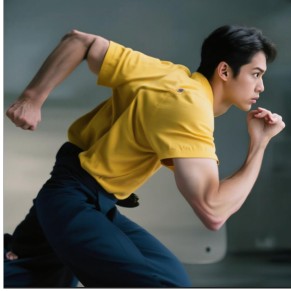

An individual is captured in a dynamic pose with their body leaning forward, their right arm bent at a 90-degree angle in front of them ...

Figure 8: Failure analysis to showcase some limitations of our method.

