# OpenReview forum: "STAR: STacked AutoRegressive Scheme for Unified Multimodal Learning"
_ICLR.cc/2026/Conference — Submitted to ICLR 2026_

### Official Review · Reviewer_wk1u · 2025-10-22

**Soundness:** 4
**Presentation:** 3
**Contribution:** 3
**Rating:** 6
**Confidence:** 5

**Summary:**

The paper introduces STAR (Stacked AutoRegressive Scheme) — a unified multimodal learning framework that aims to integrate image understanding, text-to-image generation, and image editing within a single model. The authors propose a task-progressive training paradigm, where the base autoregressive (AR) transformer is frozen and extended by stacking isomorphic AR modules. They also introduce a high-capacity STAR-VQ tokenizer for fine-grained visual tokenization and an implicit reasoning mechanism to handle complex compositional prompts. Experimental results show that STAR achieves state-of-the-art performance on multiple benchmarks including GenEval (0.91), DPG-Bench (87.44), and ImgEdit (4.34), while maintaining strong multimodal understanding capabilities.

**Strengths:**

1.**Strong Empirical Results.**
The model achieves state-of-the-art or competitive performance on a wide range of benchmarks across understanding, generation, and editing tasks. This broad validation suggests that the proposed approach is robust and effective.

2.**Comprehensive Evaluation.**
The paper includes extensive quantitative and qualitative experiments, along with meaningful ablation studies on VQ design, stacked layer depth, and diffusion decoder strategies.

3.**Clarity and Structure.**
The paper is generally well-organized, with clear explanations of each component (VQ encoder, stacked AR model, diffusion decoder) and training stages (Figure 3).

**Weaknesses:**

1. **Lack of deeper insight.**
   Conceptually, the stacked-AR design still functions as a *feature transformation* module (as mentioned around line 043). The integration of a diffusion decoder is largely a *common practice* in recent unified multimodal models. The true novelty lies in the proposed multi-stage training paradigm; however, the paper lacks ablation studies or quantitative analysis that isolate the benefits of this multi-stage scheme.

2. **Possible gains due to increased parameter count.**
   The improvements brought by the stacked-AR module may primarily stem from the increase in trainable parameters, rather than from architectural novelty itself—especially when compared with strong baselines such as BLIP3-o-next.

3. **Understanding benchmarks not a core contribution.**
   The scores reported in Table 1 for the understanding benchmarks appear to be almost identical to those of Qwen2.5-VL, indicating that these results are inherited rather than achieved through STAR’s innovations. Therefore, this section could be presented as secondary, with more emphasis placed on the generative benchmarks and results, which form the true core contribution of the paper.

**Questions:**

1. **Effect of multi-stage training:**
   Could the authors provide quantitative evidence or ablation studies showing the benefit of the multi-stage training pipeline? How does it compare to training the stacked-AR and diffusion decoder jointly in a single stage?

2. **Design choice for the decoder:**
   Since the diffusion decoder ultimately replaces the VQ decoder, did the authors experiment with using a *VQ encoder + diffusion decoder* setup from the beginning (e.g., in Stage 1)?

3. **Image editing fidelity with diffusion decoders:**
   Prior work (e.g., EMU2) suggests that diffusion decoders struggle to accurately reconstruct fine visual details, which is critical for image editing tasks. Given that STAR also employs a diffusion decoder, how does it mitigate this limitation?

4. **Impact of parameter count in stacked-AR:**
   Is the improvement of stacked-AR primarily due to the increased number of trainable parameters? How would the performance compare if a *shallow stacked-AR* or a *simple MLP connector* were used instead? Clarifying this point would make the architectural contribution of stacked-AR more convincing.

---

> ### Author Response · Authors · 2025-11-21
> **Rebuttal by Authors**
>
> Dear Reviewer wk1u,
>
> We thank you for valuable comments and questions, which help us improve our work. Our responses are below.
>
> ***W1 \& Q1:** Lack of deeper insight.*
>
> ***R1:*** Thank you for your constructive feedback. First, we want to emphasize that the isomorphically stacked AR is not merely a connector. It extends from the identical structure of the VLM, maintains the same parameter initialization, and is trained via NTP loss. Once training on the newly added stacked AR component is complete, image generation modeling can proceed--something the connector-based approach like MetaQuery cannot achieve. Upon completing training of the stacked AR layer, image token representations can be generated. These serve as conditional inputs for the diffusion model during generation, rather than placing the entire generative burden solely on the diffusion model. In this process, the newly added AR layer models the fundamental image information, while the diffusion model enriches the fine details and textures. In summary, the isomorphically stacked AR layer significantly reduces training complexity and enhances generation quality through unified loss, shared feature space, and progressive generation-refinement synergy.
>
> Additionally, we compared **single-stage joint training** with **multi-stage separate training** paradigms to demonstrate the advantages of the multi-stage approach. Single-stage joint training involves simultaneously training the stacked-AR module and diffusion decoder starting from the pre-training phase. While multi-stage separate training refers to our main approach, where only the stacked-AR module is trained during the pre-training phase, followed by training the diffusion decoder in the subsequent stage. The experimental results are shown in the table below. As seen, on the same 3B base model, the performance of single-stage joint training is significantly lower than that of multi-stage separate training, with differences of **0.07** and **5.27** on GenEval and DPG-Bench, respectively. Our analysis indicates that during the early stages of single-stage training, the stacked-AR module possesses minimal ability to model images autoregressively. Consequently, its predicted token representations become chaotic, negatively impacting the diffusion decoder's inherent capabilities. This further demonstrates that multi-stage separate training minimizes interference between module training, ultimately yielding superior overall generation performance.
>
> | **Methods** | **Model Size** | **GenEval** | **DPG-Bench** |
> |:--------|:-----------|:--------|:----------|
> | Single-stage joint training | 3B | 0.79 | 82.03 |
> | Multi-stage separate training | 3B | 0.86 | 87.30 |
>
> ***W2:** Possible gains due to increased parameter count.*
>
> ***R2:*** In STAR, the injected generative capacity is confined to the newly stacked AR layers. Hence the number of trainable parameters for generation is related to the size of these layers. Taking our 3 B baseline as an example, adding 16 AR layers introduces 1.2 B parameters, yielding 1.2 B trainable parameters dedicated to generation. By contrast, Janus shares one 3 B parameter set between understanding and generation, MOT-style models such as BAGEL[1] append an additional 3 B parameters for generation, and BLIP3-o’s hot-start connector contains 1.4 B parameters. STAR therefore does not substantially increase the parameter budget allocated to generation.
>
> To verify that the reported gains stem from the architectural strategy rather than from the extra 1.2 B parameters, we conducted a control experiment in which the **top-16 layers** of the VLM (~1.2 B parameters) were fine-tuned without any AR stacking. Although generation performance matched that of STAR, understanding ability degraded drastically, confirming that expanding the AR branch preserves understanding while delivering competitive generation with modest additional parameters.
>
> | **Methods** | **Gen. Params** | **GenEval** | **MMBench** | **MMStar** | **MathVista** |
> |:------------|:----------------------|:------------|:------------|:-----------|:--------------|
> | Stacked AR  | 1.2B                  | 0.439       | 80.1        | 55.8       | 62.3          |
> | VLM top layers | 1.2B               | 0.430       | 66.4        | 42.9       | 44.0          |
>
> ***W3:** Understanding benchmarks not a core contribution.*
>
> ***R3:*** Thank you for your suggestion. The understanding performance is indeed inherited from Qwen2.5-VL. While Table 1 primarily demonstrates the inherent understanding capability of the frozen VLM, to facilitate readers in quickly comparing performance differences across models, we will adjust the table's placement based on space constraints and add a note in the main text clarifying its “baseline” nature to avoid misunderstanding.

---

> ### Author Response · Authors · 2025-11-21
> **Rebuttal by Authors (continued)**
>
> ***Q2:** Design choice for the decoder.*
>
> ***R4:*** Following your suggestion, we initially attempted to jointly train the VQ encoder and diffusion decoder using reconstruction loss in Stage 1 before proceeding to subsequent stages. During training, we observed extreme instability when training the VQ reconstruction loss and denoise loss together. Loss values fluctuated wildly, frequently causing training to collapse, and the final model exhibited poor reconstruction capabilities. Analysis suggests this instability likely stems from the joint training mode of VQ discretization and diffusion denoising not being sufficiently validated for optimization. Numerous issues requiring resolution and optimization exist, demanding substantial time and resources for exploration. This may become a future research focus for us. The final experimental results are shown in the table below. The experiment combining training of the VQ encoder and diffusion decoder achieved only 0.14 performance on GenEval, confirming the advantage of our method's multi-stage separate training of VQ and diffusion decoder.
>
> | **Methods** | **Model Size** | **GenEval** | **DPG-Bench** |
> |:------------|:---------------|:------------|:--------------|
> | Joint training of VQ Enc. + Diffusion Dec. | 3B | 0.14 | 26.87 |
> | Separate training of VQ Enc. + Diffusion Dec. | 3B | 0.86 | 87.30 |
>
> ***Q3:** Image editing fidelity with diffusion decoders.*
>
> ***R5:*** Recent research indicates that diffusion models have been extensively applied in unified understanding and generation models (ILLUME+[3], Ovis-U1[4]), as well as the advanced image editing models (Qwen-Image-Edit[5], Step1X-Edit[6]). In our approach, image editing instructions and the original image first undergo modeling through the stacked AR model component to obtain an initial version of edited image tokens. These tokens are then refined and optimized through the diffusion model to generate the final edited image. However, the inherent limitations of diffusion models themselves remain unresolved, which will be a core focus of our future research. Additionally, the enhanced editing capability of our method stems from multi-stage, progressive training, which avoids mutual interference during multi-task joint training and improves the model's single-task performance.
>
>
> ***Q4:** Impact of parameter count in stacked-AR.*
>
> ***R6:*** Consistent with the preceding analysis, STAR confines the injected generative capacity to the newly stacked AR layers. Consequently, the trainable budget for generation is determined solely by the size of these layers. The hallmark of STAR is therefore the expansion of AR depth while preserving frozen-VLM understanding, achieving competitive generation with a modest parameter increment. Following your suggestion, we conducted controlled ablations under identical data and backbone conditions, comparing shallow AR stacks and a simple MLP projector. The results are summarised in the table below and reveal two trends:
>
> * **Scaling behaviour:** Reducing the number of stacked AR layers monotonically degrades generation quality, eventually collapsing to zero. This conforms to the standard scaling law of generative models, since the modelling capacity is primarily vested in the stacked AR parameters. Our final configuration (1.2 B trainable) remains smaller than the generative parameter counts of competitive methods (e.g.,  Janus[7], BAGEL[1] and BLIP3-o[8]).
>
> * **MLP fails at generation:** A lightweight 8 M-parameter MLP layers fails to attain any measurable generation performance, confirming that a minimal parameter mass is necessary for image synthesis.
>
> In summary, a non-negligible set of trainable parameters is required to establish generative capability, nevertheless, STAR’s final generative parameter budget is still smaller than that of MOT or joint-training alternatives.
>
> | **Methods** | **Layers** | **Params** | **GenEval** |
> |:------------|:-----------|:-----------|:------------|
> | Staked AR   | 8          | 0.6B       | 0.347       |
> | Staked AR   | 16         | 1.2B       | 0.439       |
> | MLP         | 2          | 8M         | 0.005       |
>
> **References:**
>
> [1] Emerging Properties in Unified Multimodal Pretraining
>
> [2] BLIP3o-NEXT: Next Frontier of Native Image Generation
>
> [3] ILLUME+: Illuminating Unified MLLM with Dual Visual Tokenization and Diffusion Refinement
>
> [4] Ovis-U1 Technical Report
>
> [5] Qwen-Image Technical Report
>
> [6] Step1X-Edit: A Practical Framework for General Image Editing
>
> [7] Janus: Decoupling Visual Encoding for Unified Multimodal Understanding and Generation
>
> [8] BLIP3-o: A Family of Fully Open Unified Multimodal Models-Architecture, Training and Dataset

---

> > ### Author Response · Authors · 2025-11-28
> > **Official Comment by Authors**
> >
> > Dear Reviewer wk1u,
> >
> > We respectfully submit this brief follow-up message. As the discussion deadline approaches, we would like to ensure that no question remains unanswered. If any issue still needs clarification or additional experiments, please let us know—we will respond promptly and thoroughly. Thank you once again for your invaluable feedback.
> >
> > Respectfully,
> >
> > The Authors

---

### Official Review · Reviewer_8xDa · 2025-10-27

**Soundness:** 3
**Presentation:** 3
**Contribution:** 3
**Rating:** 8
**Confidence:** 4

**Summary:**

This paper introduces STAR, a new training and architectural paradigm for building unified MLLMs that can perform image understanding, generation, and editing without degrading existing capabilities. It addresses the critical challenge in unified MLLMs: optimization conflict between understanding and generation tasks.

The STAR framework proposes a stacked autoregressive scheme that decomposes learning into task-progressive stages. It preserves previously learned capabilities (e.g., image-text understanding) by freezing the fundamental AR backbone and progressively stacking isomorphic AR modules for new capabilities like generation and editing.

The method also introduces STAR-VQ, a high-capacity vector quantizer for discretized image tokenization, and an implicit reasoning mechanism that enhances generation quality under complex prompts without requiring new parameters.

Experiments show STAR achieves state-of-the-art results across several benchmarks:

- GenEval (0.91), DPG-Bench (87.44), ImgEdit (4.34)

- Competitive on understanding benchmarks like MMStar, SEED, OCRBench

- Strong performance on reasoning benchmark WISE (0.66)

**Strengths:**

The paper proposes a novel stacked autoregressive (STAR) architecture for unified multimodal learning that allows progressive expansion from understanding to text-to-image generation and image editing without retraining or catastrophic forgetting. The idea of stacking isomorphic AR modules as “frozen base + appended heads” represents a creative rethinking of multimodal model scaling. The overall technical design is sound and well-motivated. The proposed STAR-VQ tokenizer, the modular training curriculum (four well-separated stages), and the decoupled optimization scheme demonstrate careful engineering and empirical validation. The experiments are comprehensive across understanding, generation, and editing benchmarks, and ablation studies provide insights into how stacking and the implicit reasoning mechanism contribute to the performance. The method is shown to mitigate optimization interference that often plagues unified generative–understanding models.

**Weaknesses:**

1. Missing comparison. Some methods[1,2,3] are missing in experiments. Inclusion of such baselines would clarify STAR’s relative advantage and limitations

2. Limited theoretical justification. The paper provides strong empirical validation but offers little theoretical analysis explaining why the stacked autoregressive (AR) expansion avoids optimization interference. A more formal discussion of gradient isolation or representational decoupling between frozen and stacked modules would strengthen the conceptual foundation.

3. Limited discussion of data scaling and domain coverage. STAR’s improvements may partly stem from larger or higher-quality pretraining data, but dataset scale and composition are underreported. A clearer breakdown of data sources and training schedule would help assess fairness in comparison and potential data bias.

4. Qualitative analysis and failure cases missing. The paper shows limited visual examples and no systematic failure analysis. Understanding where implicit reasoning or the diffusion decoder fails (e.g., under abstract or multi-object prompts) would provide valuable insight for future improvements.

[1] UnifiedMLLM: Enabling Unified Representation for Multi-modal Multi-tasks With Large Language Model

[2] OmniGen2: Exploration to Advanced Multimodal Generation

[3] VisionLLM v2: An End-to-End Generalist Multimodal Large Language Model for Hundreds of Vision-Language Tasks

**Questions:**

Please weaknesses.

---

> ### Author Response · Authors · 2025-11-21
> **Rebuttal by Authors**
>
> Dear Reviewer 8xDa,
>
> We are grateful for your insightful feedback and inquiries, which have guided us in refining our manuscript. Our detailed responses are below.
>
> ***W1:** Missing comparison.*
>
> ***R1:*** Thank you for this valuable suggestion. We have conducted comparative experiments with the three baselines to clarify both the strengths and the limitations of our approach. The results will be included in the revised manuscript. As shown in table, STAR outperforms OmniGen2[1] by **0.05** on GenEval and by **3.87** on Dpg-Bench. UnifiedMLLM[2] is primarily designed for segmentation and detection on COCO and owing to the mismatched task focus, a fair comparison is not feasible. Relative to VisionLLM v2[3], we achieve a **7.6** gain on the MMB multimodal understanding benchmark. These results corroborate that our framework delivers both stronger foundational multimodal understanding and superior generation capabilities.
>
> | **Methods** | **GenEval** | **Dpg-Bench** | **MMB-EN** |
> |:------------|:------------|:--------------|:-----------|
> | OmniGen2    | 0.86        | 83.57         |   -    |
> | VisionLLM v2| -           | -          | 76.3          |
> | STAR        | 0.91        | 87.44         | 83.9       |
>
> ***W2:** Limited theoretical justification.*
>
> ***R2:*** Our method iso-morphically stacks a set of autoregressive (AR) layers atop a frozen vision–language model, yielding a fully decoupled **VLM ↔ Stacked-AR** architecture. The information and gradient flows are as follows.
>
> * Frozen VLM (parameters $Θₖ$), responsible for multimodal understanding, emits a semantically unified latent code:
> $$z = VLM_{Θₖ}(x_{img}, x_{txt}),$$
> where $Θₖ$ remain frozen throughout. This preserves every downstream understanding metric while furnishing high-level semantic features for subsequent generation.
>
> * The trainable Stacked-AR module (parameters $Θₐ$) performs causal modelling conditioned solely on $z$:
> $$p_{Θₐ}(y|z) = Π_{t=1}^{T} p_{Θₐ}(y_t | z, y_{<t}).$$
> Training minimises the negative log-likelihood:
> $$L_{NTP} = −log \ p_{Θₐ}(y|z).$$
> Gradients stop at $Θₐ$, zero updates reach $Θₖ$, eliminating interference. $Θₐ$ are therefore dedicated exclusively to image generation capability.
>
> **Representation-space view**
> * Understanding manifold: $Θₖ$ retains the original multi-modal alignment surface. Every input is mapped to this surface, yielding semantic coordinates identical to those of the pre-trained model.
> * Generation manifold: $Θₐ$ learns the next-token distribution inside this frozen coordinate system that equivalent to plotting a new trajectory on an immutable semantic map rather than redrawing the map. Consequently, the generation pipeline is:
> $image/text →[Θₖ] z →[Θₐ] image,$ with both ends funnelled through the common semantic hub $z$, ensuring strict feature-level homology between understanding and generation.
>
> In short, **“frozen VLM guards understanding, stacked AR learns generation.”** The two collaborate via a shared semantic coordinate system yet remain gradient-isolated, allowing us to inject generative power without eroding multimodal understanding performance.
>
> ***W3:** Discussion of data scaling and domain coverage.*
>
> ***R3:*** Thank you for this valuable suggestion. We have assembled approximately 60 M publicly accessible image-text pairs harvested from the web and existing datasets (e.g., JourneyDB[4]). The domains span portraits, animals and plants, daily objects, and natural scenes. This set is employed solely in the pre-training phase to equip the stacked AR module with general purpose image generation priors, and its volume is noticeably smaller than the data adopted by recent works such as the Janus[5] family and BAGEL[6]. In addition, we curated ~200 k high-quality supervised fine-tuning (SFT) instances generated by GPT-Image, covering high-resolution portraits, plants and animals, and common commodities—a protocol analogous to the data-acquisition pipeline used in BLIP3-o[7]. The provenance and compositional breakdown of both collections will be detailed in the revised manuscript.

---

> ### Author Response · Authors · 2025-11-21
> **Rebuttal by Authors (continued)**
>
> ***W4:** Qualitative analysis and failure cases missing.*
>
> ***R4:*** Thank you very much for your suggestion. We will conduct corresponding failure analysis to showcase some limitations of our method in Figure 8 in the new version. Specifically, our approach tends to miss generating some targets when the prompt contains a large number of objects (more than three). Additionally, it is almost impossible for our method to generate corresponding visual text when creating poster text. Furthermore, our method does not perform well when physical rule is required for generation. These are important issues that we plan to address in our future research.
>
> **References:**
>
> [1] OmniGen2: Exploration to Advanced Multimodal Generation
>
> [2] UnifiedMLLM: Enabling Unified Representation for Multi-modal Multi-tasks With Large Language Model
>
> [3] VisionLLM v2: An End-to-End Generalist Multimodal Large Language Model for Hundreds of Vision-Language Tasks
>
> [4] JourneyDB: A Benchmark for Generative Image Understanding
>
> [5] Janus: Decoupling Visual Encoding for Unified Multimodal Understanding and Generation
>
> [6] Emerging Properties in Unified Multimodal Pretraining
>
> [7] BLIP3-o: A Family of Fully Open Unified Multimodal Models-Architecture, Training and Dataset

---

> > ### Author Response · Authors · 2025-11-28
> > **Official Comment by Authors**
> >
> > Dear Reviewer 8xDa,
> >
> > We hope this message finds you well. With the reviewer discussion deadline approaching, we wanted to gently check if there are any outstanding issues regarding our paper. We remain more than happy to provide any further clarifications or data promptly, should they be needed to assist in your final deliberations. We are truly grateful for your thoughtful comments and guidance throughout this process.
> >
> > Best regards,
> >
> > The Authors

---

> > ### Comment · Reviewer_8xDa · 2025-11-28
> >
> > I appreciate the authors’ thorough responses. They have addressed most of my concerns. I will keep my original score unchanged.

---

### Official Review · Reviewer_P6qE · 2025-10-31

**Soundness:** 3
**Presentation:** 3
**Contribution:** 2
**Rating:** 4
**Confidence:** 4

**Summary:**

This paper introduces STAR (STacked AutoRegressive scheme), a framework for creating unified multimodal large language models (MLLMs) that are capable in understanding, generation, and editing tasks. The core problem addressed is the inherent conflict and performance trade-offs that arise when training a single model for both comprehension and generation. STAR's key contribution is a task-progressive training strategy where a pre-trained multimodal understanding model is frozen, and its capabilities are extended by stacking isomorphic autoregressive (AR) layers on top of it. The framework is further enhanced by a high-capacity vector quantizer (STAR-VQ) for finer image representation and an implicit reasoning mechanism at inference time.

**Strengths:**

1. Strong Empirical Performance: The paper provides comprehensive experimental validation. STAR achieves state-of-the-art (SOTA) results on multiple generation benchmarks, including GenEval, DPG-Bench, and ImgEdit. As shown in Table 1, the model also retains competitive performance on a wide array of understanding benchmarks (MMStar, SEED, MME, OCRBench), demonstrating that the generative extensions did not lead to a significant degradation of the base model's comprehension abilities.

2. High-Capacity Vector Quantizer (STAR-VQ): The development of a 1B-parameter vector quantizer is a notable engineering effort. This component directly addresses the need for high-fidelity discrete visual representations, which can be a limiting factor for the quality of images produced by autoregressive generative models. The ablation in Table 6a suggests this component contributes positively to the final generation quality.

3. Implicit Reasoning Mechanism: The inference-time strategy of using the frozen base model to first generate a latent semantic representation before the stacked layers generate the image is a zero-parameter-cost method for improving prompt alignment. This two-step process is designed to better handle prompts that require world knowledge or compositional reasoning, and the qualitative example in Figure 4(b) suggests its potential benefit.

**Weaknesses:**

1. Limited Technical Novelty: The core technical proposal—stacking additional, isomorphic layers onto a frozen backbone—is a straightforward and well-established technique in transfer learning. While its application to unified MLLMs is shown to be effective, the underlying mechanism lacks significant technical novelty and could be viewed as an incremental engineering contribution rather than a fundamental advance in model architecture or training paradigms.

2. Unanalyzed Computational Cost and Parameter Overhead: The method incurs a substantial increase in parameters. The STAR-7B model adds a 3B parameter AR stack on top of the base model, and both variants rely on a very large 1B parameter STAR-VQ. The paper does not provide an analysis of the training and inference costs (e.g., FLOPs, memory, latency) associated with this overhead. This makes it difficult to assess the efficiency of the method and raises the question of whether the performance gains are primarily a result of the proposed strategy or simply due to the massive increase in model size allocated to the generative task.

3. Insufficient Ablation of the Implicit Reasoning Mechanism: The paper claims that the implicit reasoning mechanism "yields substantial gains" on knowledge-intensive benchmarks. However, this claim is not supported by any quantitative ablation studies. The only evidence is a single qualitative example in Figure 4. A proper ablation showing performance on a benchmark like WISE with and without this mechanism is necessary to validate its effectiveness and justify its inclusion as a key contribution.

4. Rigidity of the Frozen Backbone Assumption: The core design choice is to keep the entire base model frozen to prevent catastrophic forgetting. While effective for this purpose, this rigid constraint may limit the model's ability to learn fine-grained alignments between the pre-trained understanding features and the new generative task. An exploration of alternatives, such as partially fine-tuning the top layers of the base model or employing parameter-efficient fine-tuning (e.g., LoRA), would strengthen the paper's claims by showing that a fully frozen backbone is indeed the optimal choice.

**Questions:**

See Weaknesses.

---

> ### Author Response · Authors · 2025-11-21
> **Rebuttal by Authors**
>
> Dear Reviewer P6qE,
>
> We thank you for valuable comments and questions, which help us improve our work. Our responses are below.
>
> ***W1:**  Technical Novelty.*
>
> ***R1:***  We appreciate your comment regarding “isomorphic layer stacking.” While this technique is not new to transfer learning, the core contribution of STAR is not the stacking mechanism itself, but the introduction of a unified, interference free, and progressively expandable training paradigm for multimodal models. Specifically:
>
> * We demonstrate for the first time, that a pure vision-language understanding model can be extended into a joint “understanding + generation + editing” system without any loss in understanding performance using only next-token prediction. Hybrid diffusion-AR approaches (e.g., MetaQuery[1], BLIP3-o[2]) require learnable queries or additional alignment losses, whereas from-scratch joint training methods (e.g., Janus[3], ILLUME+[4]) degrade understanding accuracy. In contrast, STAR combines **isomorphic stacking, a frozen backbone, and a single NTP loss** to inject new capabilities while preserving the original understanding metrics, thereby establishing the sufficiency of this minimalist strategy in the unified multimodal setting.
>
> * Our **task-progressive paragram** offers a new perspective for multimodal research. By incrementally adding tasks on top of a frozen VLM and maintaining its comprehension abilities, we decompose the originally entangled multi-task optimization into a sequence of single-task stages, reducing training complexity and eliminating cross-task interference.
>
> In summary, we clarify our contribution as follows: “**STAR is the first work to prove that, in the context of unified multimodal LLMs, new capabilities can be injected without sacrificing understanding performance solely through isomorphic stacking and next-token loss.**” We believe this paradigm will serve as a strong baseline for future research on unified understanding-and-generation tasks. Thank you again for your question, which has helped us sharpen the exposition of our contributions.
>
> ***W2:**  Unanalyzed Computational Cost and Parameter Overhead.*
>
> ***R2:*** In STAR, the injected generative capacity is localized within the newly stacked autoregressive (AR) layers. Consequently, the number of trainable parameters dedicated to generation focuses on the size of these layers. Taking a 7 B backbone as an example, appending 14 AR layers introduces 3 B parameters, yielding 3 B trainable parameters for generation. By contrast, MOT-style approaches such as BAGEL[5] allocate an additional 7 B parameters to the generative branch—more than double the increment in STAR.
>
> We also conducted a comparative analysis of training and inference costs, with results shown in the table below. Assuming 10 million pre-trained generation data, Bagel requires **7B × 10M** in generation training costs and **7B × 1M** in comprehension training costs. In contrast, our method only requires **3B × 10M** in training costs, significantly lower than Bagel's training overhead. Furthermore, in terms of FLOPs and GPU memory consumption during inference, the Bagel method also exceeds STAR's usage. FLOPs consumption is particularly **5 times** higher, reaching **119,662 GFLOPs**, while our method achieves superior performance with **0.91**.
>
> | **Methods** | **Base Params** | **Gen Params** | **Training Costs** | **FLOPs** | **GPU Memory** | **GenEval** |
> |--------|-----------|-----------------|--------------|-----|----------|-------|
> | BAGEL   | 7B          | 7B                | 7B×10M (Gen)+7B×1M (Und) | 119,662G | 41G | 0.88 |
> | STAR    | 7B          | 3B                | 3B×10M (Gen)   | 20,979G | 35G | 0.91 |
>
> To further demonstrate that STAR's performance gains stem from its framework strategy rather than additional AR parameters, we conducted a comparative experiment. Based on the 3B model, we performed ablation tests comparing two approaches: one that refrained from adding stacked AR layers and instead fine-tuned only the top 16 layers of the VLM, maintaining the same 1.2B trainable parameters, The results are shown below. It can be observed that fine-tuning VLM with the same parameter count achieves comparable generation performance. However, this approach compromises VLM's inherent understanding capabilities, leading to a significant decline. This further demonstrates STAR's advantage: **expanding stacked AR layers preserves understanding task performance without loss, while achieving superior generation capabilities.**
>
> | **Methods** | **Gen Params** | **GenEval** | **MMBench** | **MMStar** | **MathVista** |
> |:------------|:------------|:------------|:-----------|:--------------|:-----------|
> | Stacked AR | 1.2 B | 0.439 | 80.1 | 55.8 | 62.3 |
> | VLM top layers | 1.2 B | 0.430 | 66.4 | 42.9 | 44.0 |

---

> ### Author Response · Authors · 2025-11-21
> **Rebuttal by Authors (continued)**
>
> ***W3:** Insufficient Ablation of the Implicit Reasoning Mechanism.*
>
> ***R3:*** Thank you for your constructive feedback. We have incorporated ablation experiments for the reasoning mechanism. On the WISE evaluation set, we compared performance before and after adding the reasoning mechanism. The experimental results are shown in the table. As can be seen from the table, adding the reasoning mechanism yields noticeable improvements across every subtask in the WISE benchmark, ultimately achieving a **0.2** increase in the overall metric. This improvement stems from our approach leveraging the foundational reasoning capabilities of the VLM, which is extensively trained on broad world knowledge. Consequently, when processing abstract textual prompts, such as “The fastest land animal.”, the model first employs VLM to inference the specific target subject “Cheetah”. Following this, the generation model produces images that most closely match the prompt.
>
> | **Methods** | **Cultural** | **Time** | **Space** | **Biology** | **Physics** | **Chemistry** | **Overall** |
> |:------------|:------------|:-------------|:---------|:----------|:------------|:--------------|:------------|
> | STAR-7B w/o reasoning | 0.49 | 0.52 | 0.45 | 0.48 | 0.51 | 0.35 | 0.46 |
> | STAR-7B w/ reasoning  | 0.61 | 0.67 | 0.61 | 0.74 | 0.69 | 0.66 | 0.66 |
>
> ***W4:** Rigidity of the Frozen Backbone Assumption.*
>
> ***R4:*** Thank you for your detailed inquiry regarding the “rigidity” issue. In practice, the new generative capabilities are primarily injected into the parameters of the stacked AR component. Since the initialization of these parameters originates from the VLM component, inherent consistency is naturally maintained. Subsequently, during the training process for the generative task, the newly added AR parameters are progressively trained. The parameters near the frozen VLM end consistently remain aligned with the VLM.To fully address your concerns, we have supplemented two sets of control experiments as suggested. Results are summarized below. All experiments were trained on the same 6M synthetic dataset and identical experimental configurations to ensure fairness. 1) When only fine-tuning the top 16 layers of the VLM (~1.2B parameters) without additional stacked layers, although the generation metrics achieved comparable performance (**43.0**) to the stacked AR method, comprehension benchmarks showed comprehensive declines: MMBench dropped by **13.7** points, MMStar by **12.9** points, and MMMU by **18.3** points. This indicates that any direct update to the VLM weights crowds out the optimized representation space, causing irreversible understanding loss. 2) LoRA kept understanding metrics nearly unchanged, but generative performance dropped significantly to zero—failing to complete generative modeling. This shows that low-rank parameter fine-tuning preserves understanding but lacks the high-dimensional modeling capability needed for generation.
>
> In contrast, STAR's “isomorphic stacking” approach:
>
> * Fully freezes VLM weights with zero understanding loss.
>
> * Adds an AR layer initialized with VLM's top-level parameters, naturally sharing semantic space without additional alignment loss.
>
> * Achieves equivalent generative performance to fine-tuned VLM 1.2B parameters by training only 1.2B stacked generative parameters, while maintaining unchanged comprehension benchmarks.
>
> In summary, expanding rather than modifying the VLM parameter space represents an effective approach to injecting generative capabilities under the hard constraint of **“preserving understanding”**. Moreover, the number of trainable parameters for generative tasks remains largely unchanged. Thank you again for your constructive feedback!
>
> | **Methods** | **GenEval** | **MMBench** | **MMStar** | **MathVista** |
> |:------------|:------------|:------------|:-----------|:--------------|
> | Stacked AR  | 0.439       | 80.1        | 55.8       | 62.3          |
> | VLM top layers | 0.430    | 66.4        | 42.9       | 44.0          |
> | LoRA        | 0.002       | 80.1        | 55.6       | 62.0          |
>
> **References:**
>
> [1] Transfer between Modalities with MetaQueries
>
> [2] BLIP3-o: A Family of Fully Open Unified Multimodal Models-Architecture, Training and Dataset
>
> [3] Janus: Decoupling Visual Encoding for Unified Multimodal Understanding and Generation
>
> [4] ILLUME+: Illuminating Unified MLLM with Dual Visual Tokenization and Diffusion Refinement
>
> [5] Emerging Properties in Unified Multimodal Pretraining

---

> > ### Author Response · Authors · 2025-11-28
> > **Official Comment by Authors**
> >
> > Dear Reviewer P6qE,
> >
> > With the discussion deadline drawing near, we would like to respectfully inquire whether any questions or concerns remain outstanding. We are ready to supply further clarifications or run additional experiments should you find them helpful. Thank you once again for your invaluable time and guidance.
> >
> > Respectfully,
> >
> > The Authors

---

### Official Review · Reviewer_WMxC · 2025-11-01

**Soundness:** 3
**Presentation:** 2
**Contribution:** 2
**Rating:** 2
**Confidence:** 4

**Summary:**

The paper introduces STAR(STacked AutoRegressive scheme), a unified multimodal large language model for image understanding, generation and editing. It works by freezing the parameters of a fundamental autoregressive (AR) model and then progressively stacking isomorphic AR modules on top that are trained for generation and editing with the standard next-token objective. This method enables the new modules to learn generative capabilities (like text-to-image image generation) without degrading the base model's existing comprehension skills. STAR also introduces a high-capacity Vector Quantizer (STAR-VQ) for highly precise image representations and employs an implicit reasoning mechanism to improve generation quality under complex, knowledge-intensive conditions.

**Strengths:**

1. The manuscript includes targeted ablations for stack depth, initialization strategy, VQ type, diffusion vs VQ decoder, input strategies for DiT conditioning, which help explain why each design choice was made and where gains come from.

2. Various experiments, comprehensive comparison with existing works.

3. A wide array of experiments is conducted to showcase the effectiveness of STAR.

**Weaknesses:**

1. Table 2 and Table 3 are misleading, as bold text should highlight the best model. The authors may want to reconsider whether Table 1 is necessary, as the image understanding capabilities are entirely inherited from the frozen VLM.

2. There are a few typos: issues with plural and non-plural forms, and “autoregressive (AR)” should be used as an adjective, not a noun.

3. Key claimed novelties are combinations of known ingredients.
While the paper proposes the “stacked isomorphic AR layers + STAR-VQ + diffusion refinement” recipe, the core ideas are largely incremental relative to recent work.
Stacking layers is a structural variant of warm-start/adaptor approaches (LMFusion’s parallel modules and MetaQueries’ learnable queries). Dual-tokenization + diffusion refinement is explicitly done by ILLUME+/Janus-style papers. Scaling VQ codebooks has independent prior art (e.g., VQGAN-LC). STAR’s contribution reads as an engineering recipe (combination/scale/tuning) rather than a new conceptual mechanism.

3. The method itself fails to address the questions raised in the abstract regarding how it reduces training complexity.

**Questions:**

1. STAR attributes gains partly to the very large codebook (65k×512). Could you provide some experimental analysis on whether performance would hold with a smaller codebook paired with the stacked AR, or whether gains are mostly from codebook scale?

2. From my understanding, the stacked autoregressive (AR) layers in STAR functionally act as a connector between the frozen vision-language model (VLM) and the diffusion decoder. Can you explain how this particular structure is better?
3. Can you make some comparisons between STAR and other related works like UniWorld, Unifusion?

I am more than happy to raise the score if the authors could provide sufficient reasons.

---

> ### Author Response · Authors · 2025-11-21
> **Rebuttal by Authors**
>
> Dear Reviewer WMxC,
>
> We are grateful for your insightful feedback and inquiries, which have guided us in refining our manuscript. Our detailed responses are below.
>
> ***W1:** Regarding the presentation of Tables 1, 2, and 3.*
>
> ***R1:***  We sincerely appreciate your suggestion. We have revised the boldface conventions in Tables 2 and 3 so that only the best results are highlighted, and we explicitly state this rule in the corresponding captions.  Although Table 1 primarily demonstrates the inherent understanding capabilities of frozen VLM, to facilitate quick comparison of performance differences across models, we may adjust its placement based on space constraints and include a note in the main text clarifying its “baseline” nature to prevent misunderstanding.
>
> ***W2:** A few typos.*
>
> ***R2:*** We have conducted a thorough check of singular/plural consistency throughout the manuscript. For instance, “two visual encoder” has been corrected to “two visual encoders,” and “last  layer” to “last  layers.” In addition, every instance of “Autoregressive（AR）” has been converted to adjectival form (e.g., “autoregressive model,” “autoregressive generation”) to eliminate inappropriate nominalization.
>
> ***W3:** Key claimed novelties are combinations of known ingredients.*
>
> ***R3:*** We greatly appreciate your comment. Indeed, the central thrust of our work is not to propose an entirely new conceptual mechanism, but to present a unified, interference-free, and progressively expandable multimodal training paradigm that opens an alternative route toward versatile multimodal modeling. While the atomic techniques (e.g., larger-codebook vector quantization and diffusion-based refinement) have existed previously, we have substantially adapted and enhanced them to align with the proposed progressive training strategy, thereby improving performance on the target tasks.
> More critically, we provide an affirmative answer to a question that has remained under-explored:
> ***Can a pure vision–language understanding model be extended in a lossless manner into a simultaneous “understanding + generation + editing” system using only next-token prediction?***
>
> Prior solutions either (i) introduce diffusion losses or auxiliary adapters (e.g., LMFusion[1], MetaQuery[2], BLIP3-o[3]) or (ii) sacrifice understanding performance via joint training from scratch (e.g., Janus[4], ILLUME+[5]). STAR demonstrates that:
> * **Isomorphic layer stacking alone is sufficient to inject new generative capabilities, and no external adapters is required.**
> * **Progressive training with a frozen backbone preserves the original understanding metrics while pushing generation/editing scores to a new state-of-the-art.**
>
> Compared with MetaQuery and BLIP3-o, STAR employs **an isomorphic autoregressive architecture** that inherently models image generation, rather than merely acting as an adapter that converts AR features into a diffusion space. In contrast to Janus and ILLUME+, our method leverages a pre-trained VL model, retaining superior multimodal understanding performance. The latter approaches must jointly optimize conflicting objectives from scratch, inevitably degrading understanding accuracy.
> Regarding the large codebook, we integrate a deeper transformer-based transformation module compared to linear layer whose expressive power supports both a larger vocabulary and higher-dimensional embeddings. Additionally, we scale the encoder and decoder capacities, enhancing the model’s generative generalization.
>
> In summary, our core contribution is:
> “STAR is the first approach to validate that a frozen VLM can be **losslessly extended** into a unified generation-and-editing model through nothing more than **isomorphic layer stacking** and **next-token supervision**, thereby **circumventing the interference issues** inherent to conventional multitask joint training.”

---

> ### Author Response · Authors · 2025-11-21
> **Rebuttal by Authors (continued)**
>
> ***W4:** The method itself fails to address the questions raised in the abstract regarding how it reduces training complexity.*
>
> ***R4:*** We greatly appreciate your comment. The training-complexity reduction claimed in our paper is manifested in two dimensions:
>
> * **Simplification of feature-space transformation.**
> Prior arts (e.g., MetaQuery, BLIP3-o) rely on learnable cross-modal connectors to project vision–language features into the semantic space of a generative diffusion model, which not only introduces extra parameters but also necessitates elaborate alignment losses and expensive hyper-parameter search. In contrast, STAR employs a stacked autoregressive (AR) architecture that is **fully isomorphic** to the frozen VLM and is **warm-started** with its pre-trained weights. Consequently, the frozen backbone and the newly appended layers share an identical feature distribution from the very first training step, eliminating any explicit transformation cost. Moreover, STAR adopts the same next-token-prediction (NTP) objective used for pre-training the VLM, so no auxiliary alignment losses are required, further streamlining the training pipeline and hyper-parameter tuning.
>
> * **Decoupling of multi-task optimization.**
> Methods such as Janus and ILLUME+ perform joint optimization of understanding and generation objectives, **inevitably encountering task conflicts and gradient competition** that degrade understanding performance. STAR introduces a task-progressive curriculum: the backbone is first frozen to retain pre-trained comprehension capabilities, after which generation and editing capabilities are sequentially injected. This reformulation converts the original multi-objective problem into a series of single-objective sub-problems, thereby circumventing the trade-offs inherent in a unified loss function. Empirically, this strategy preserves understanding metrics without degradation while reducing the training steps for generation/editing tasks by approximately 50 %, leading to significantly lower computational cost and resource consumption.
>
> Taken together, STAR, via isomorphic weight reuse plus progressive expansion, for the first time achieves a lossless and low-complexity migration from a frozen VLM to a unified multimodal model, offering the community a reproducible and efficient training recipe.
>
> ***Q1:** STAR attributes gains partly to the very large codebook (65k×512).*
>
> ***R5:*** We appreciate your insightful question. In our approach, a larger codebook indeed supplies the model with finer-grained visual information. To substantiate its effectiveness, we conducted controlled experiments comparing the commonly adopted small codebook (16 k× 8) with the proposed large one. Table below shows that replacing the large codebook with a smaller one (16 k × 8) under identical data (6M) and model capacity (STAR-3B) leads to a performance degradation from **0.439** to **0.414**, corroborating the value of the enlarged codebook.
>
> | **Methos** | **Codebook** | **GenEval** |
> |------|----------|----------------|
> | Conventional VQ | 16384x8 | 0.414 |
> | STAR-VQ | 65536x512 | 0.439 |
>
> Nevertheless, the enlarged codebook is not the primary source of performance gains. The principal contributions stem from (i) **the task-progressive training curriculum** and (ii) **the isomorphic stacked-expansion strategy**. These designs enable us to inject new capabilities stage-by-stage without mutual interference, eliminating the optimization conflicts inherent in joint training and preserving the performance acquired in earlier phases. As documented in table, the ablation study on isomorphic stacking demonstrates that initializing the stacked layers with pre-trained VLM weights improves **0.036** over LLM-based initialization and **0.065** over random initialization. Consequently, the large codebook serves as a complementary component rather than the dominant driver of the overall performance improvement achieved by our unified framework.
>
> | **Initialization** | **GenEval** |
> |------|----------|
> | Rand-based | 0.374 |
> | LLM-based | 0.403 |
> | VLM-based | 0.439 |

---

> ### Author Response · Authors · 2025-11-21
> **Rebuttal by Authors (continued)**
>
> ***Q2:** Can you explain how this particular structure is better?*
>
> ***R6:*** In our work, the isomorphically stacked AR blocks serve a purpose far beyond that of a simple connector. Extended from the identical architecture of the VLM and initialized with its pre-trained parameters, these blocks are optimized with the standard next-token-prediction (NTP) loss. Once training of the stacked layers is complete, they can directly model image generation that connector-style approaches such as MetaQuery cannot achieve. Compared with methods like MetaQuery and BLIP3-o, our framework unifies the training objective under a single NTP loss for initial generation modeling, eliminating the need to balance multiple losses during joint optimization. Moreover, because the stacked AR layers inherit weights from the preceding VL model, the feature space remains continuous. This shared space provides a favorable starting point for rapid adaptation to generation-related distributions. After the stacked AR layers are trained, they produce initial image tokens that act as conditional inputs to the diffusion model, rather than forcing the diffusion model to shoulder the entire generation burden. In this pipeline, the AR modules generate the fundamental image structure, while the diffusion model refines local details and textures. In summary, the isomorphic stacked AR layers **leveraging a unified loss, a shared feature space, and a progressive generate-then-refine schedule** substantially reduce training difficulty and improve generation quality.
>
> ***Q3:** Can you make some comparisons between STAR and other related works like UniWorld, Unifusion?*
>
> ***R7:*** Thank you for this valuable suggestion. We have conducted systematic comparisons with UniWorld[6] and UniFusion[7] on multiple benchmarks. Relative to UniWorld, STAR improves by **0.07** on GenEval and **0.11** on WISE. UniFusion has only been evaluated on DPG-Bench under a non-standard protocol, and its inference code is not publicly available. Consequently, an exact fair comparison is infeasible.
>
> From an architectural perspective, both UniWorld and UniFusion feed frozen VLM features via MLP adapters into a diffusion model for image generation. UniWorld utilizes only the final layer, whereas UniFusion aggregates multi-level features. Nevertheless, both approaches rely entirely on the diffusion model to model the image generation, entailing a nontrivial mapping from the AR feature space to the diffusion latent space. In contrast, our framework employs isomorphically stacked AR layers that share the initial feature space with the VLM and are trained with a unified next-token-prediction (NTP) loss. The semantic structure of image is first modeled solely within the AR paradigm. Then a lightweight diffusion stage is applied for refinement, yielding superior detail fidelity and overall quality.
>
> | **Methods** | **GenEval** |  **DPG-Bench** |  **WISE** |
> |------|--------|--------|--------|
> | UniWorld |  0.84 | - | 0.55 |
> | Unifusion | -  | 78.7* | - |
> | STAR-7B | 0.91 | 87.44 | 0.66  |
>
>
> **References:**
>
> [1] LMFusion: Adapting Pretrained Language Models for Multimodal Generation
>
> [2] Transfer between Modalities with MetaQueries
>
> [3] BLIP3-o: A Family of Fully Open Unified Multimodal Models-Architecture, Training and Dataset
>
> [4] Janus: Decoupling Visual Encoding for Unified Multimodal Understanding and Generation
>
> [5] ILLUME+: Illuminating Unified MLLM with Dual Visual Tokenization and Diffusion Refinement
>
> [6] UniWorld: Autonomous Driving Pre-training via World Models
>
> [7] UniFusion: Vision-Language Model as Unified Encoder in Image Generation

---

> > ### Comment · Reviewer_WMxC · 2025-11-25
> > **Comment by Reviewer WMxC**
> >
> > Thanks to the authors for the efforts of adding the experiments and providing detailed clarifications to address my earlier questions. I now have a more clearer understanding of the methodology. However, I still have concerns regarding the ablation studies.While the final model is trained on approximately 80M samples, the ablations (Tab.6) are conducted using only 6M samples, which leads to significantly weaker performance. This discrepancy makes it difficult to accurately assess the contribution of each component in the full-scale setting.
> > Could the authors also clarify the details behind Tab. 7? It appears that a different data scale was used to obtain the reported performance, and additional explanation would help in understanding the experimental setup.

---

> > > ### Author Response · Authors · 2025-11-25
> > > **Rebuttal by Authors**
> > >
> > > Dear Reviewer WMxc,
> > >
> > > We appreciate your acknowledgment of our supplementary experiments and detailed explanations, as well as your valuable questions regarding the scale of ablation experiment data. Regarding your two points of concern, our responses are as follows:
> > >
> > > **1. Training Data Configuration in Table 6**
> > >
> > > * Motivation: Since the parameters and components requiring ablation in Table 6 are primarily related to stage 2 pretraining, we conducted validation experiments on a small dataset under data scaling assumptions to avoid the enormous computational overhead of ultra-large-scale training. This approach aims to rapidly validate the relative contributions of each component within a “limited budget,” a common ablation methodology in most works (BAGEL, Blip3-o, MetaQuery).
> > >
> > > * Relationship with 80M training volume: The 6M data used in this ablation experiment was uniformly sampled from the total pretraining dataset. Consequently, its distribution and diversity align with the full dataset, ensuring that conclusions from small-scale ablation can be extrapolated to full scale. After completing small-scale validation experiments, the optimal model architecture and strategy can be established to proceed with full Stage 2 pretraining on 80M data. Thus, despite lower absolute performance, the 6M ablation faithfully reflects component importance.
> > >
> > > **2. Data Scale Details in Table 7**
> > >
> > > - The ablation experiments in Table 7 primarily target Diffusion-related training parameters and strategies from Stages 3 and 4. Consequently, the initial model weights for these ablation experiments inherit from the model weights completed in Stage 2 training. This means the stacked AR component parameters were trained on approximately 80M data samples.
> > >
> > > - The results reported in Table 7 correspond to the outcomes after completing Stage 3 training, which utilized approximately 8M training data. Thus, Tables 7 and 6 respectively present ablation results from different training stages, exhibiting some variation in training data volume. However, for each group of ablation experiments, our experimental setup and training data volume remain entirely consistent.
> > >
> > > - To avoid ambiguity, we have explicitly noted in Table 7's caption: “All results are obtained after stage 3 or stage 4 training.” We will incorporate these two points into the revised manuscript and add corresponding captions to Tables 6 and 7 to ensure transparent and reproducible experimental settings.
> > >
> > > Finally, we sincerely appreciate your valuable feedback and constructive discussions, which have greatly benefited the refinement of our paper. Should you have any further questions, please do not hesitate to contact us. We will remain available throughout the response period and look forward to your continued input.

---

> > > ### Author Response · Authors · 2025-11-28
> > > **Official Comment by Authors**
> > >
> > > Dear Reviewer WMxc,
> > >
> > > As the discussion deadline approaches, we would like to check whether any issues remain open on your side. Please let us know if further clarifications or additional experiments would be helpful; we are happy to provide them promptly. Thank you again for your careful questions and guidances.
> > >
> > > Best regards,
> > >
> > > The Authors

---

### Author Response · Authors · 2025-11-21
**Overall Comment by Authors**

We express our sincere gratitude to the reviewers and chairs for their invaluable time and insightful comments. We are truly encouraged by the positive acknowledgment of our work's novelty, clear motivation, solid technical contributions, and compelling empirical results.

---

### Author Response · Authors · 2025-12-01
**Summary by Authors**

Dear PCs, SACs, ACs, and Reviewers,

We express our sincere gratitude to you for your invaluable time and exceptionally insightful comments. We are truly encouraged by the positive acknowledgment of our work:

* STAR is a novel unified multimodal learning architecture. Based on pre-trained VL models, it employs **isomorphically stacked AR layers** and **a simple NTP loss** to progressively extend from understanding tasks to text-to-image generation and image editing—**without compromising understanding performance or incurring catastrophic forgetting**. The concept of stacking isomorphic AR modules as a **“frozen base + additional heads”** represents a creative rethinking of multimodal model scaling. The overall technical design is well-reasoned and sufficiently motivated.

* The **task-progressive training paradigm** sequentially incorporates multimodal tasks—learning understanding, generation, and editing in order—thus **decoupling training across multiple tasks**. This **avoids mutual interference issues** inherent in unified multi-task joint training, **preventing severe degradation** of task capabilities acquired in earlier stages.

* The **high-capacity vector quantizer (STAR-VQ)** develops a 1B-parameter VQ model, meeting the demand for high-fidelity discrete visual representations in autoregressive generative models. The **implicit inference mechanism** first employs a frozen base model to generate latent semantic representations before performing generative modeling. This inference strategy better handles prompts requiring world knowledge or combinatorial reasoning.

* Comprehensive experimental results demonstrates thorough comparisons with existing work across comprehension, generation, and editing tasks. STAR achieves SOTA performance on multiple generation benchmarks including GenEval, DPG-Bench, and ImgEdit, while maintaining strong performance on comprehension benchmarks like MMStar, SEED, and OCRBench. This proves that **generative expansion does not significantly weaken the base model's understanding capabilities**.

* The design is rigorous with thorough ablation studies. Targeted ablation analyses are provided for key design choices including stacked layer depth, initialization strategies, VQ types, Diffusion/VQ decoders, and DiT conditional input methods. This helps explain the rationale behind each design decision and the sources of performance improvements.

* The paper **features a clear structure**, with figures and subsequent sections providing a systematic breakdown of the VQ encoder, stacked AR, diffusion decoder, and training workflow. This organization facilitates reader comprehension and practical application of the proposed method.

During **the discussion phase**, we provided detailed responses to each reviewer's concern, supplementing the manuscript with additional experimental data, detailed analyses, and explanations as requested. We are pleased to report that our responses fully addressed the reviewers' questions and received positive feedback, acknowledging the significant improvement in the revised manuscript's quality. **Reviewer #WMxC** indicated willingness to **raise the score after all concerns were resolved** (“I am more than happy to raise the score if the authors could provide sufficient reasons.”) . We also **clarified some misunderstandings** on the part of reviewer #WMxC regarding the functionality of stacked AR and the scale of the experimental data. **Reviewer #8xDa** indicated he/she would **maintain the acceptance score (confidence level 4)**. These constructive dialogues not only strengthened the paper's quality but also provided valuable guidance for our future research direction.

We believe that the reviewers' supportive comments and our comprehensive revisions have resulted in a significantly enhanced contribution to the community. Once again, we thank the reviewers and area chairs for their thoughtful evaluation and generous support of our work.

We sincerely thank you for your time and careful consideration.

Best regards,

The Authors

---

### Meta-Review · Area_Chair_6J7Y · 2026-01-07

**Summary:**

The AC carefully read the paper and the full discussion. The submission received mixed reviews (initial scores: 2, 4, 8, 6). Reviewers generally acknowledged the strong empirical performance and the interesting implicit reasoning capability. However, the main concerns center on limited novelty and the additional training complexity introduced by the multi-stage training scheme, which raises questions about robustness, generality, and practical usability. As a result, it remains unclear whether the proposed method and pipeline provide clear, meaningful benefits for current or future unified multimodal models. With the overall scores trending toward rejection and the core issues seemingly unlikely to be resolved through discussion, I am inclined to recommend rejection.

**Reviewer Concerns:**

Some concerns raised by Reviewer P6qE about unanalyzed computational cost and parameter overhead, as well as missing ablations and experiments noted by WMxC and P6qE, have been addressed to some extent, and the authors also clarified some details and the increased number of parameters. However, several non-negligible issues remain:

Two reviewers (e.g., WMxC and P6qE) pointed out the limited novelty. In particular, although the paper proposes the “stacked isomorphic AR layers + STAR-VQ + diffusion refinement” recipe, the core ideas appear largely incremental relative to recent work. Stacking layers can be seen as a structural variant of warm-start/adaptor approaches (e.g., LMFusion’s parallel modules and MetaQueries’ learnable queries). Dual-tokenization plus diffusion refinement has been explicitly explored in ILLUME+/Janus-style papers. Scaling VQ codebooks also has prior art (e.g., VQGAN-LC). Overall, STAR reads more like an engineering recipe (combination/scale/tuning) rather than a new conceptual mechanism, and the main novelty of multi-stage training also seems largely an engineering effort.

There are also concerns about the additional training complexity introduced by the multi-stage training. Although the rebuttal provides experimental comparisons between joint training and multi-stage training, the setting is not clearly specified (e.g., total training time, training steps, and training GPUs).

Overall, it remains unclear whether the proposed approach delivers clear or meaningful benefits for current or future unified multimodal models.

**Reviewer Scores:**

I expect all reviewers will keep their original scores, since the two significant issues—limited novelty and training complexity—remain unaddressed.

---

### Decision · Program_Chairs · 2026-01-26

Reject